# ATTRIBUTING RESPONSE TO CONTEXT: A JENSEN–SHANNON DIVERGENCE DRIVEN MECHANISTIC STUDY OF CONTEXT ATTRIBUTION IN RETRIEVAL-AUGMENTED GENERATION

**Ruizhe Li**[1]* **Chen Chen**[2] **Yuchen Hu**[2] **Yanjun Gao**[4] **Xi Wang**[5] **Emine Yilmaz**[3]
[1]University of Aberdeen  [2]Nanyang Technological University  [3]University College London
[4]University of Colorado Anschutz Medical Campus  [5]University of Sheffield

## ABSTRACT

Retrieval-Augmented Generation (RAG) leverages large language models (LLMs) combined with external contexts to enhance accuracy and reliability of generated responses. However, reliably attributing generated content to specific context segments, *context attribution*, remains challenging due to computationally intensive nature of current methods, which often require extensive fine-tuning or human annotation. In this work, we introduce a novel **J**ensen–**S**hannon **D**ivergence driven method to **A**ttribute **R**esponse to **C**ontext (**ARC-JSD**), enabling efficient and accurate identification of essential context sentences without additional fine-tuning, gradient-calculation or surrogate modelling. Evaluations on a wide range of RAG benchmarks, such as TyDi QA, Hotpot QA, and Musique, using instruction-tuned LLMs in different scales demonstrate superior accuracy and significant computational efficiency improvements compared to the previous baselines. Furthermore, our mechanistic analysis reveals specific attention heads and multilayer perceptron (MLP) layers responsible for context attribution, providing valuable insights into the internal workings of RAG models and how they affect RAG behaviours. Code is available at https://github.com/ruizheliUOA/ARC_JSD.

## 1 INTRODUCTION

Retrieval-Augmented Generation (RAG), leveraging LLMs, has demonstrated significant potential in both academic research (Qian et al., 2024; Yue et al., 2025; Song et al., 2025) and industrial applications (Yang et al., 2024; Wang et al., 2025a; Guo et al., 2025) by enhancing the accuracy and grounding of generated responses through external contexts such as provided documents or retrieved articles online. A key benefit of RAG lies in its ability to mitigate the hallucination by explicitly attributing generated responses to specific segments of the provided context, known as *context attribution*[1] (Wang et al., 2024; Qi et al., 2024; Cohen-Wang et al., 2024; Chuang et al., 2025).

Nevertheless, verifying the extent to which generated responses are genuinely grounded in their cited context remains a challenging task. Current approaches frequently rely heavily on human annotation (Zeng et al., 2021; Menick et al., 2022; Slobodkin et al., 2024) or computationally expensive methods such as model fine-tuning and gradient-based feature attribution for accurate attribution (Yue et al., 2023; Qi et al., 2024; Chuang et al., 2025), particularly when dealing with extensive documents. For instance, Qi et al. (2024) utilised distribution shifts between responses generated with and without context to identify relevant tokens and employed gradient-based feature attribution to pinpoint context relevance. Similarly, Chuang et al. (2025) enhanced context attribution accuracy through reward-driven fine-tuning within a Direct Preference Optimisation (DPO) framework, based on probability drop and hold analysis of model outputs to context ablation.

---

* Corresponding Author: ruizhe.li@abdn.ac.uk

[1]We use the term *context attribution* in this work, and there are several different terms used in this area, such as citation, self-citation, etc.

To circumvent these computationally intensive methods, Cohen-Wang et al. (2024) introduced an inference-time attribution mechanism premised on the assumption that if removing grounded context segments substantially reduces the probability of a generated response, those segments are deemed necessary. Conversely, if retaining only grounded segments maintains response probability, these segments are considered sufficient. By capturing hundreds of probability ablation variations per context-response pair, they trained a linear surrogate model based on those hundreds of vectors, including the context segment masks and the corresponding generation probability of the original response, to identify context segments crucial for grounding model responses.

However, Cohen-Wang et al. (2024) still needs hundreds of RAG model's forward calls to collect probability ablation samples for the linear surrogate model training. We propose a novel inference-time **J**ensen–**S**hannon **D**ivergence driven method to **A**ttribute **R**esponse to **C**ontext (ARC-JSD), building upon the inference-attribution assumption above. Our method evaluates the divergence in response distributions generated under the full context compared to sentence-ablated contexts, ranking context sentences based on their JSD differences because of JSD's symmetric, finite, scale-free, and bounded properties (see § 3 and § 8 for details). This approach offers a significant computational advantage, as it eliminates the need for any additional fine-tuning or surrogate modelling. Furthermore, our ARC-JSD can avoid missing or smoothing non-linearities using JSD to directly quantify actual output distribution shift compared to the linear surrogate modelling (Cohen-Wang et al., 2024).

We empirically evaluate our JSD-driven context attribution approach across multiple question-answering benchmarks, i.e., TyDi QA (Clark et al., 2020), Hotpot QA (Yang et al., 2018), and MuSiQue (Trivedi et al., 2022), using state-of-the-art instruction-tuned LLMs including Qwen2-1.5B-Instruct, Qwen2-7B-Instruct (Yang et al., 2024), Gemma2-2B-Instruct, and Gemma2-9B-Instruct (Team et al., 2024). Our results not only demonstrate improved average accuracy over 10% in context attribution but also achieve computational efficiency, achieving up to a three-fold speedup compared to Cohen-Wang et al. (2024)'s linear-surrogate-based and other gradient-based baselines.

Moreover, we investigate deeper into a mechanistic exploration of context attribution within RAGs by integrating JSD-based analysis with Logit Lens (nostalgebraist, 2020). Through systematic probing, we identify specific attention heads and MLP layers critical for context attribution. By subsequently manipulating these located attention heads and MLPs using JSD scores as confidence gates, we can control RAGs and further mitigate the hallucination rate. In addition, we can visualise how relevant knowledge is stored in the corresponding MLP layers.

In summary, our primary contributions include:

1. Developing a JSD-driven context attribution method that accurately identifies context critical for grounding responses without requiring fine-tuning, surrogate modelling or gradient calculation.
2. Proposing a computationally efficient solution that can be readily integrated into any existing RAGs. Conducting a detailed mechanistic analysis of RAG, systematically uncovering and validating attention heads and MLP layers responsible for context attribution behaviours.
3. Mitigating the hallucination rate and controlling RAG behaviour by manipulating located attention heads and MLPs within RAG.

## 2 RELATED WORK

**Context attribution for RAG.** Prior works for context attribution mainly focus on teaching RAG LLMs to generate self-citations for responses, such as few-shot in-context learning (Gao et al., 2023), instruction fine-tuning (Ye et al., 2024). Some post-hoc works (Chen et al., 2023; Qi et al., 2024) used an auxiliary language model or gradient-based feature attribution to locate relevant context segments. In general, those methods for context attribution are *corroborative* (Worledge et al., 2024) in nature, as citations within context are evaluated on whether they *support* or *imply* a generated response. Meanwhile, Cohen-Wang et al. (2024); Chuang et al. (2025); Cohen-Wang et al. (2025); Liu et al. (2024) including our work focus on the *contributive* attribution methods, which are used to identify whether citations *cause* RAG LLMs to generate a response. Chuang et al. (2025) proposed a reward-based fine-tuning with DPO to guide RAG LLMs for context attribution, and Cohen-Wang et al. (2024; 2025) further trained a linear surrogate model to identify context segments crucial for grounding model responses. Liu et al. (2024) focuses on formalising and comparing different attribution acceleration methods and ignores attribution accuracy improvements. However, compared

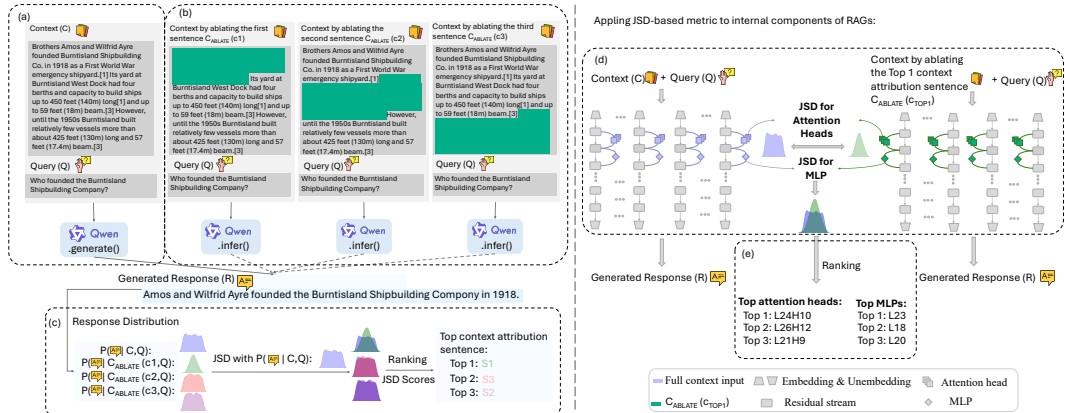

Figure 1: This framework demonstrates how our ARC-JSD works: *(a)* a RAG LLM $\mathcal{P}_{LM}(\cdot)$ first generates response $\mathcal{R}$ conditioned on full context $\mathcal{C}$ and query $\mathcal{Q}$ input; *(b)* By ablating single context sentence once a time, we can calculate probability distribution of the same response $\mathcal{R}$ conditioned on the ablated context $\mathcal{C}_{ABLATE}(c_i)$ and query $\mathcal{Q}$; *(c)* We further calculate JSD scores about probability distribution of the same response $\mathcal{R}$ conditioned on full context and ablated context, and locate the most relevant context sentence supporting $\mathcal{R}$ with the highest JSD score. Then, we apply JSD-based metric to internal components of RAGs: *(d)* For each attention head or MLP output at each layer, we calculate probability distribution of the same response $\mathcal{R}$ conditioned on the same query $\mathcal{Q}$ with full context $\mathcal{C}$ and ablated context $\mathcal{C}_{ABLATE}(c_{top\text{-}1})$ by removing top relevant context sentence based on § 4.1; *(e)* We can further locate top-$N$ relevant attention heads or MLPs which contribute the context attribution by ranking the collected JSD scores with a descending order.

to Cohen-Wang et al. (2024); Chuang et al. (2025); Cohen-Wang et al. (2025) and corroborative methods above, our ARC-JSD method eliminates the need for any additional fine-tuning or surrogate modelling, and it can be directly integrated into any existing RAGs.

**Mechanistic analysis for RAG.** Existing mechanistic studies focus on next token generation task to analyse internal mechanisms of attention heads or MLPs, such as hallucination detection (Ferrando et al., 2025), multiple-choice questions (Li & Gao, 2024; Wiegreffe et al., 2025; Wang et al., 2025b) and knowledge editing (Meng et al., 2022a; 2023; Katz et al., 2024). Recently, Sun et al. (2025) used a mechanistic interpretability method to analyse attention heads and MLPs of RAGs for the hallucination detection task. Compared to Sun et al. (2025) focusing on locating sources which lead to hallucinations, our proposed ARC-JSD can be regarded as a complementary method to locate citations within context segments and analyse attentions and MLPs, which *causes* RAG LLMs to generate a correct response. Wu et al. (2025a) focuses on mechanistically analysing retrieval attention heads of RAG LLMs under the Needle-in-the-Haystack (NIAH) setting, where they mainly evaluate whether retrieval attention heads conduct a copy-and-paste operation for retrieving a semantically irrelevant "needle" sentence from the context to the model's outputs. Compared to Wu et al. (2025a), which restricts their mechanistic analysis to the NIAH setting where the model performs copy-and-paste retrieval, our work investigates how RAG LLMs mechanistically generate responses based on retrieved content through paraphrasing and contextual integration. This setting better reflects real-world RAG applications[2], where models rarely copy text exactly but instead synthesise and rephrase information from retrieved sources.

## 3 BACKGROUND

**Problem Setup.** Consider an autoregressive Transformer-based language model (LLM), denoted as $\mathcal{P}_{LM}(\cdot)$. Under RAG settings, this model generates responses ($\mathcal{R}$) based on an input query ($\mathcal{Q}$) and associated context ($\mathcal{C}$). Formally, response generation process can be described as $\mathcal{R} \sim \mathcal{P}_{LM}(\cdot|\mathcal{C}, \mathcal{Q})$, where context $\mathcal{C}$ consists of sentences $(c_1, c_2, \ldots, c_{|\mathcal{C}|})$, the query $\mathcal{Q}$ comprises tokens $(q_1, q_2, \ldots, q_{|\mathcal{Q}|})$, and the generated response $\mathcal{R}$ includes tokens $(r_1, r_2, \ldots, r_{|\mathcal{R}|})$. Our analysis

---

[2]Compared to traditional RAG to directly map prompts based on their word embeddings, our work has a more general setting, which avoids potential embedding mismatch due to the common paraphrase of RAGs.

of context attribution focuses on how the entire response distribution changes when conditioned on the full context set and ablated context alongside the query: $\mathcal{R} \sim \mathcal{P}_{\text{LM}}(\cdot|c_1, \ldots, c_{|\mathcal{C}|}, \mathcal{Q})$, $\mathcal{R} \sim \mathcal{P}_{\text{LM}}(\cdot|\mathcal{C}_{\text{ABLATE}}(c_i), \mathcal{Q})$ where $\mathcal{C}_{\text{ABLATE}}(c_i) = \mathcal{C} \setminus \{c_i\}$, $i \in \{1, \ldots, |\mathcal{C}|\}$.

**Logit Lens.** Logit lens (nostalgebraist, 2020) is a mechanistic interpretability method designed to analyse intermediate representations within autoregressive Transformers. Given the LLM architecture described in Appendix D, logit lens leverages intermediate representations to quantify the direct contribution of attention heads $(\mathbf{a}_i^{\ell,h})$, MLP outputs $(\mathbf{m}_i^{\ell})$, and residual streams $(\mathbf{x}_i^{\ell})$ to token logits: $\text{logit}_i^{\ell,h}(\mathbf{a}_i^{\ell,h}) = W_U\sigma(\mathbf{a}_i^{\ell,h}), \text{logit}_i^{\ell}(\mathbf{m}_i^{\ell}) = W_U\sigma(\mathbf{m}_i^{\ell}), \text{logit}_i^{\ell}(\mathbf{x}_i^{\ell}) = W_U\sigma(\mathbf{x}_i^{\ell})$. Thus, logit lens serves as a powerful tool for pinpointing specific model components crucial to prediction behaviours.

**JSD for Context Attribution.** JSD is a symmetrised, smoothed variant of Kullback–Leibler (KL) divergence that quantifies information gap (in bits) between two probability distributions. Because it is symmetric, finite, scale-free, and bounded in $[0, \log 2]$, JSD allows scores from different layers to be compared directly, without sensitivity to arbitrary logit shifts. Following "logit lens" perspective of Sun et al. (2025), we treat JSD as model's belief of how much its next-token distribution will change. Concretely, we compute JSD between the full-context token distribution and the distribution obtained after removing a single retrieved sentence $c_i$. A high divergence indicates that the model's internal representations (and therefore its output logits) depend strongly on $c_i$. Empirically, ablating the sentence with the highest JSD causes the largest drop in answer likelihood, validating JSD as a concise and reliable signal for context attribution in RAG models (See § 8 for comparisons among JSD, Wasserstein, Total Variation (TV) and Maximum Mean Discrepancy (MMD)).

## 4 Attributing Top Relevant Context Sentences via JSD

### 4.1 Identifying Relevant Context via JSD

Following assumption proposed by Cohen-Wang et al. (2024), removal of context segments critical to generating a specific response $\mathcal{R}$ significantly impacts the probability distribution of that response. Conversely, the removal of less relevant context segments is expected to minimally affect $\mathcal{R}$.

Unlike the approach by Cohen-Wang et al. (2024), which requires extensive sampling of ablated contexts for each $(\mathcal{C}, \mathcal{Q})$ pair and training a surrogate model to learn context-response relationships, our proposed ARC-JSD method relies purely on inference in the Fig. 1. Specifically, we compute the JSD between the response probability distributions conditioned on the full context $\mathcal{C}$ and on each context-ablated variant $\mathcal{C}_{\text{ABLATE}}(c_i)$:

$$\text{JSD}(c_i) = \sum_{j=1}^{|\mathcal{R}|} \text{JSD}\left(\mathcal{P}_{\text{LM}}(r_j|\mathcal{C}, \mathcal{Q})||\mathcal{P}_{\text{LM}}(r_j|\mathcal{C}_{\text{ABLATE}}(c_i), \mathcal{Q})\right) \tag{1}$$

where we use $\text{JSD}(c_i)$ to aggregate the JSD score of each generated tokens $r_j$ from $\mathcal{R}$ when the context sentence $c_i$ is ablated from the context $\mathcal{C}$. By calculating JSD scores for all sentences in the context, we identify the most relevant context sentence $c_i$ by selecting the sentence based on the assumption about the significant impact of removing critical context segments: $c_{\text{Top-1}} = \arg\max_{c_i \in \mathcal{C}} \left(\{\text{JSD}(c_i)\}_{i=1}^{|\mathcal{C}|}\right)$.

### 4.2 Evaluation of Context Attribution Accuracy

To assess efficacy of our ARC-JSD method, we conduct experiments on three widely recognised question-answering datasets commonly used in RAG studies: *TyDi QA* (Clark et al., 2020): a multilingual QA dataset using entire Wikipedia articles as external context (we only use English part), *Hotpot QA* (Yang et al., 2018): a multi-hop QA dataset requiring reasoning for questions based on multiple documents, and *MuSiQue* (Trivedi et al., 2022): a high-quality multi-hop QA benchmark over Wikipedia that highlights minimal context and multiple valid

Table 1: The FLOPs for each baseline and ARC-JSD, where $P$ indicates the number of target model parameters, $T$ indicates the number of tokens per context sentence, $L$ is layer numbers of target model, $|\mathcal{R}|$ and $|\mathcal{C}|$ indicates the number of response tokens and context sentences, respectively.

| Baselines | Theoretical FLOPs | Slowdown over ARC-JSD |
|---|---|---|
| ALTI-Logit | $2PT|\mathcal{C}||\mathcal{R}|L$ | $|\mathcal{R}|L/|\mathcal{C}|$ |
| MIRAGE | $4PT|\mathcal{C}|(2|\mathcal{C}| + 1)$ | $4 + 2/|\mathcal{C}|$ |
| Contextcite (32 calls) | $2PT \times 32^2$ | $(32/|\mathcal{C}|)^2$ |
| Contextcite (256 calls) | $2PT \times 256^2$ | $(256/|\mathcal{C}|)^2$ |
| ARC-JSD | $2PT|\mathcal{C}|^2$ | 1 |

reasoning paths to evaluate complex reasoning capabilities. Moreover, we evaluate our ARC-JSD with different training-free baselines for context attribution: *ALTI-Logit* (Ferrando et al., 2023): a method to directly compare logit difference between input context and generation on token level by accumulating layerwise logit; *MIRAGE* (Qi et al., 2024): a gradient-based and token-level method to locate context-sensitive tokens using contrastive feature attribution; *Contextcite* (Cohen-Wang et al., 2024): a post-hoc method to train a linear surrogate model based on a fixed group of context ablation forward runs. Table 2 summarises the statistics of these datasets, where *MuSiQue* has the longest context input compared to others, with the average length of context in sentences $|\mathcal{C}| = 93.6$. Our evaluations involve four instruction-tuned LLMs of varying scales, namely Qwen2-1.5B-IT, Qwen2-7B-IT (Yang et al., 2024), Gemma2-2B-IT, and Gemma2-9B-IT (Team et al., 2024). For each dataset, we randomly select up to 1,000 samples from its development sets. All models are evaluated in inference mode without further fine-tuning. We mainly evaluate the top-1 context attribution accuracy, which indicates the percentage of overlap between the predicted top-1 context sentence and gold-standard sentence on the datasets[3]. For ALTI-Logit and MIRAGE, which mainly focus on token-level attribution, we use the accumulated operations to locate sentence-level context attribution prediction (Appendix E includes more details).

Table 1 lists theoretical floating-point operations (FLOPs) for each method, where we follow the assumption from Kaplan et al. (2020); Hoffmann et al. (2022), i.e., one forward pass needs approximately $2PT$ FLOPs. ARC-JSD is considerably cheaper than baselines because it pinpoints salient context sentences without back-propagation or iterative token masking. ContextCite requires a fixed 32 or 256 forward passes; this is economical only when input contains more than 32 or far more than 256 sentences, respectively, but its context attribution accuracy remains below that of ARC-JSD (see Fig. 2(a)). Fig. 2(a) presents compute-accuracy trade-off on

Table 2: The size of three benchmarks randomly sampled from their development dataset is up to 1000, where the average word numbers and sentence numbers of context (i.e., $|\mathcal{C}|$) are summarised.

| Datasets | Size | Contexts | |
| --- | --- | --- | --- |
| | | Avg. Words | Avg. Sents. |
| TyDi QA | 440 | 99.5 | 4.8 |
| Hotpot QA | 1,000 | 940.3 | 51.1 |
| MuSiQue | 1,000 | 1753.8 | 93.6 |

MuSiQue dataset across all baselines and LLM backbones. It clearly demonstrates that ARC-JSD consistently outperforms all baselines, yielding an average context attribution accuracy improvement of approximately 10.7%. Although Contextcite-32 is more efficient when $|\mathcal{C}|$ is larger than 32, its attribution accuracy lags behind ARC-JSD. Overall, our method offers substantial computational efficiency improvements, achieving up to 3-fold speedups and consistently aligns with Pareto-optimal over multiple orders of magnitude for different LLM backbones. In addition, we utilise GPT-4.1 mini as a judge to compare whether generated responses of all RAG models are semantically equivalent to the corresponding gold answers from datasets when context attribution is correct. The average accuracy is up to 99.3% (See Appendix H and F for details).

## 5 MECHANISTICALLY STUDY RAG LLMS FOR CONTEXT ATTRIBUTION

**Locating Relevant Attention Heads and MLPs.** To better understand internal mechanisms by which RAG LLMs attribute generated responses to their relevant context sentences, we systematically investigate specific attention heads and MLP layers involved. Our method combines the ARC-JSD metric described previously (§ 4.1) with the Logit Lens (nostalgebraist, 2020) to precisely quantify contributions from these internal model components.

Following the ARC-JSD framework in the § 4.1, we apply JSD difference at the level of individual attention heads and MLP layers in Fig. 1, comparing their outputs between scenarios involving full context and the ablation of the most relevant context sentence using Eq. 1:

$$\text{JSD}_{\text{Attn}}^{\ell,h} = \sum_{j=1}^{|\mathcal{R}|} \text{JSD}\left(\mathcal{P}_{\text{Attn}}^{\ell,h}(r_j|\mathcal{C},\mathcal{Q})||\mathcal{P}_{\text{Attn}}^{\ell,h}(r_j|\mathcal{C}_{\text{ABLATE}}(c_{\text{top-1}}),\mathcal{Q})\right)$$

$$\text{JSD}_{\text{MLP}}^{\ell} = \sum_{j=1}^{|\mathcal{R}|} \text{JSD}\left(\mathcal{P}_{\text{MLP}}^{\ell}(r_j|\mathcal{C},\mathcal{Q})||\mathcal{P}_{\text{MLP}}^{\ell}(r_j|\mathcal{C}_{\text{ABLATE}}(c_{\text{top-1}}),\mathcal{Q})\right)$$

(2)

---

[3]We choose sentence level because current QA datasets only have sentence-level gold labels to evaluate attribution accuracy. However, our ARC-JSD method can be extended to finer-grained interactions such as phrases or sub-sentence spans by dynamically selecting the start and end token indices.

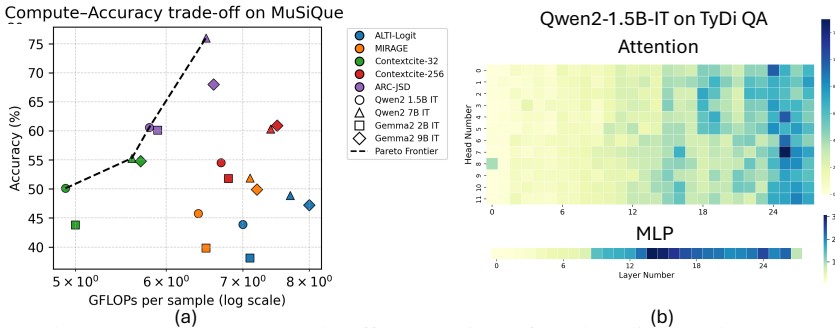

Figure 2: *(a)* The compute-accuracy trade-off on MuSiQue for 4 baselines and ARC-JSD on 4 LLM backbones with GFLOPs $\log_{10}$ scale per sample; *(b)* The average JSD score of attention heads and MLP of Qwen2-1.5B-IT on TyDi QA across all layers. The deeper colour indicates larger JSD scores.

where $\mathcal{P}^{\ell,h}_{\text{Attn}}()$ and $\mathcal{P}^{\ell}_{\text{MLP}}()$ denote the probability distributions derived from attention head outputs $\mathbf{a}^{\ell,h}_j$ and MLP outputs $\mathbf{m}^{\ell}_j$, respectively, via the logit lens and softmax operations:

$$\mathcal{P}^{\ell,h}_{\text{Attn}}() = \text{Softmax}(\text{logit}(\mathbf{a}^{\ell,h}_j)), \quad \mathcal{P}^{\ell}_{\text{MLP}}() = \text{Softmax}(\text{logit}(\mathbf{m}^{\ell}_j)) \quad (3)$$

where the shape of attention head output $\mathbf{a}^{\ell,h}$ and MLP output $\mathbf{m}^{\ell}$ is $[1, d]$, and $d$ is dimensionality of residual stream. By computing JSD scores across all heads and MLP layers, we rank these components according to their relevance to context attribution:

$$J_{\text{Top-}N}(\text{Attn}) = \text{sort}\left(\{\text{JSD}^{\ell,h}_{\text{Attn}}\}^{L,H}_{\ell=0,h=0}, \text{descending}\right), J_{\text{Top-}N}(\text{MLP}) = \text{sort}\left(\{\text{JSD}^{\ell}_{\text{MLP}}\}^{L}_{\ell=0}, \text{descending}\right) \quad (4)$$

**Mechanistic Insights from Located Attention Heads and MLPs.** Applying methodology described above, we conducted experiments across three benchmark datasets (see § 4.2) using various LLM scales. Fig. 2(b) presents distribution and JSD scores of attention heads identified as most relevant for context attribution in Qwen2-1.5B-Instruct on TyDi QA dataset. Our analysis reveals that the top attention heads contributing to context attribution predominantly reside in the higher layers. This observation holds across most datasets, partially corroborating earlier findings by Wu et al. (2025a), which indicated that retrieval-related attention heads are typically found in the intermediate and higher layers. Notably, our work expands upon NIAH setting explored by Wu et al. (2025a) by mechanistically evaluating attention heads and MLPs relevance through paraphrasing and contextual integration of RAG LLMs. This setting better reflects real-world RAG applications, where models rarely copy text exactly but instead synthesise and rephrase information from retrieved sources. Additional visualisations and distributions for another Qwen2-7B-IT and Gemma2 models across all datasets are provided in Appendix K. Similarly, Fig. 2(b) illustrates that intermediate and higher MLP layers also significantly contribute to context attribution. This pattern remains consistent across different datasets and model scales within the same LLM family. Corresponding detailed findings for Qwen2-7B-IT and Gemma2 models are available in Appendix K.

## 6 VERIFICATION OF JSD-BASED MECHANISTIC STUDY

Having identified critical RAG components using JSD in § 5, we now seek to verify these findings from a different analytical angle. Our JSD approach measures a component's dependence on context through ablation. As a crucial cross-check, we introduce the semantic gain metric, which measures a component's direct contribution to the correct answer when the full context is present. This allows us to test a key hypothesis: components that are truly important for attribution should not only depend on the context (high JSD) but also actively use it to improve the answer (high semantic gain). Demonstrating a strong correlation between these metrics will robustly verify that the components we located are indeed the primary drivers of context attribution.

### 6.1 SEMANTIC GAINS OF ATTENTION AND MLPs FOR CONTEXT ATTRIBUTION

Apart from locating relevant attention heads and MLPs using JSD-based metric from the § 5, we also know that semantic information of context attribution from attentions and MLPs will be added back to the residual stream from each layer based on the autoregressive language model's architecture

from the § 3 (Elhage et al., 2021; Katz et al., 2024). Based on such properties, we can verify whether the JSD-based metric for attention and MLPs location in the § 5 works by projecting the residual stream before and after each layer's attention and MLPs components into the vocabulary space, and calculating the cosine similarity with the generated response $\mathcal{R}$ to further identify which attention and MLP modules provide higher semantic gains.

Based on the introduction of internal mechanism of LLMs in the § 3 and full context $\mathcal{C}$ with query $\mathcal{Q}$ as model's input, we further split the residual stream flow of each layer into three parts for each generated token $t_i$, i.e., pre-residual stream $\mathbf{x}_i^{\ell,\text{pre}}$, middle-residual stream $\mathbf{x}_i^{\ell,\text{mid}}$ and post-residual stream $\mathbf{x}_i^{\ell,\text{post}}$: $\mathbf{x}_i^{\ell,\text{pre}} = \mathbf{x}_i^{\ell-1,\text{post}}, \mathbf{x}_i^{\ell,\text{mid}} = \mathbf{x}_i^{\ell,\text{pre}} + \mathbf{a}_i^\ell, \mathbf{x}_i^{\ell,\text{post}} = \mathbf{x}_i^{\ell,\text{mid}} + \mathbf{m}_i^\ell = \mathbf{x}_i^{\ell+1,\text{pre}}$. After applying the logit lens to $\mathbf{x}_i^{\ell,\text{pre}}, \mathbf{x}_i^{\ell,\text{mid}}$ and $\mathbf{x}_i^{\ell,\text{post}}$ via the softmax, we will have the probability distribution of the generated token $t_i^{\ell,\text{pre}}, t_i^{\ell,\text{mid}}$ and $t_i^{\ell,\text{post}}$ for each layer, and then we will use greedy decoding to select the top-1 token with the highest probability: $t_i^{\ell,\text{pre/mid/post}} = \arg\max_{t_i^{\ell,\text{pre/mid/post}} \in \mathcal{V}} \left( \text{softmax}\left(\text{logit}(\mathbf{x}_i^{\ell,\text{pre/mid/post}})\right)\right)$. Consequently, we can project the selected token $t_i^{\ell,\text{pre/mid/post}}$ into the vocabulary embedding space via the unembedding matrix $W_U \in \mathbb{R}^{d \times |\mathcal{V}|}$: $\mathbf{e}_i^{\ell,\text{pre/mid/post}} = W_U[:t_i^{\ell,\text{pre/mid/post}}]$. We can calculate the corresponding semantic gains $\Delta_i^{\ell,\text{Attn}}$ and $\Delta_i^{\ell,\text{MLP}}$ via attention and MLP modules using the cosine similarity difference with the generated response token embedding $\mathbf{e}_i = W_U[:r_i]$: $\Delta_i^{\ell,\text{Attn}} = \cos(\mathbf{e}_i^{\ell,\text{mid}}, \mathbf{e}_i) - \cos(\mathbf{e}_i^{\ell,\text{pre}}, \mathbf{e}_i), \Delta_i^{\ell,\text{MLP}} = \cos(\mathbf{e}_i^{\ell,\text{post}}, \mathbf{e}_i) - \cos(\mathbf{e}_i^{\ell,\text{mid}}, \mathbf{e}_i)$. Finally, we will average across the entire generated responses $\mathcal{R}$ and calculate the semantic gains $\Delta^{\ell,\text{Attn}}$ and $\Delta^{\ell,\text{MLP}}$ for attention MLP of each layer, and collect and sort the semantic gains of attention and MLP from all layer with descending order:

$$\Delta^{\ell,\text{Attn}} = \frac{1}{|\mathcal{R}|}\sum_i^{|\mathcal{R}|} \Delta_i^{\ell,\text{Attn}}, \quad \Delta^{\ell,\text{MLP}} = \frac{1}{|\mathcal{R}|}\sum_i^{|\mathcal{R}|} \Delta_i^{\ell,\text{MLP}} \tag{5}$$

$$G_{\text{Top-}N}(\text{Attn}) = \text{sort}\left(\{\Delta^{\ell,\text{Attn}}\}_{\ell=0}^L, \text{descending}\right), \quad G_{\text{Top-}N}(\text{MLP}) = \text{sort}\left(\{\Delta^{\ell,\text{MLP}}\}_{\ell=0}^L, \text{descending}\right) \tag{6}$$

## 6.2 MUTUALLY VERIFYING JSD-BASED MECHANISTIC STUDY VIA THE SEMANTIC GAINS OF ATTENTION AND MLPs

Based on the Eq. 4 and Eq. 6, we can locate layer-wise attention and MLP components relevant to context attribution from two different perspectives in the § 5 and § 6.1. We can evaluate the correlation of both metrics and further verify the effectiveness of our proposed ARC-JSD metric in the § 4.1 and § 5.

Given $\{\text{JSD}_{\text{MLP}}^\ell\}_{\ell=0}^L$ and $\{\Delta^{\ell,\text{MLP}}\}_{\ell=0}^L$ via the JSD-based and Semantic-gain-based metrics, we first define an average-ranking fusion, called *consensus* $S^{(+)}$, to fuse both JSD and semantic gain views, which is based on the assumption that a layer is important if both metrics sort the layer highly:

$$S^{(+)} = \frac{1}{2}\left(\text{ranking}_J + \text{ranking}_G\right) = \frac{1}{2}\left(\frac{\text{ranking of }\left(\{\text{JSD}_{\text{MLP}}^\ell\}_{\ell=0}^L\right)}{L} + \frac{\text{ranking of }\left(\{\Delta^{\ell,\text{MLP}}\}_{\ell=0}^L\right)}{L}\right) \tag{7}$$

where ranking of $(\cdot)$ will assign 1 to the largest $\text{JSD}_{\text{MLP}}^\ell$ or $\Delta^{\ell,\text{MLP}}$ and the smallest $\text{JSD}_{\text{MLP}}^\ell$ or $\Delta^{\ell,\text{MLP}}$ will be assigned $L$. Then we uniform and remove the layer influence divided by $L$ to get $\text{ranking}_J$ and $\text{ranking}_G$, whose range is $[1/n, 1]$, i.e., a smaller fraction will have a higher ranking ($1/n$ is best). Finally, we take the average of the $\text{ranking}_J$ and $\text{ranking}_G$ as the *consensus* $S^{(+)}$, where a smaller consensus inside of $S^{(+)}$ will indicate a stronger joint evidence that both metrics consider the layer important, and a larger consensus means at least one metric puts the layer far down the list. Finally, we calculate Spearman $\rho$ of $J_{\text{Top-}N}(\text{MLP}) \cap S_{\text{Top-}N}^{(+)}$ and $G_{\text{Top-}N}(\text{MLP}) \cap S_{\text{Top-}N}^{(+)}$, where $S_{\text{Top-}N}^{(+)} = \text{sort}(S^{(+)}, \text{ascending})$. For attention components, we first average JSD scores of all attention heads in the same layer to build $\{\text{JSD}_{\text{Attn}}^\ell\}_{l=0}^L = \{\frac{1}{H}\sum_{h=0}^H \text{JSD}_{\text{Attn}}^{\ell,h}\}_{l=0}^L$, and then further calculate $\rho$ of $J_{\text{Top-}N}(\text{Attn}) \cap S_{\text{Top-}N}^{(+)}$ and $G_{\text{Top-}N}(\text{Attn}) \cap S_{\text{Top-}N}^{(+)}$. The benefit of using *consensus* $S^{(+)}$ instead of the raw JSD or semantic gain values is that $S^{(+)}$ will remove all scaling issue due to the different units and variances of JSD or semantic gains, and a single extremely large JSD or semantic gain will not swamp the fusion, which is robust to outliers.

Table 3: Spearman's $\rho$ of the overlap about top-10 located attentions and MLPs between JSD-based mechanistic and semantic gain-based metrics over all datasets and RAG models. $\diamondsuit$ and $\spadesuit$ indicate $p$-value is $< 0.05$ and $< 0.01$, respectively.

| Top-10 Modules | Datasets | Qwen2 1.5B IT | | Qwen2 7B IT | | Gemma2 2B IT | | Gemma2 9B IT | |
|---|---|---|---|---|---|---|---|---|---|
| | | $J(\cdot) \cap S^{(+)}$ | $G(\cdot) \cap S^{(+)}$ | $J(\cdot) \cap S^{(+)}$ | $G(\cdot) \cap S^{(+)}$ | $J(\cdot) \cap S^{(+)}$ | $G(\cdot) \cap S^{(+)}$ | $J(\cdot) \cap S^{(+)}$ | $G(\cdot) \cap S^{(+)}$ |
| Attention | TyDi QA | 6.83$\diamondsuit$ | 7.26$\diamondsuit$ | 6.91$\diamondsuit$ | 7.31$\diamondsuit$ | 7.62$\spadesuit$ | 7.25$\diamondsuit$ | 7.63$\spadesuit$ | 7.28$\diamondsuit$ |
| | Hotpot QA | 6.73$\diamondsuit$ | 6.65$\diamondsuit$ | 6.81$\diamondsuit$ | 6.79$\diamondsuit$ | 6.68$\diamondsuit$ | 6.67$\diamondsuit$ | 6.72$\diamondsuit$ | 6.73$\diamondsuit$ |
| | MuSiQue | 6.67$\diamondsuit$ | 6.72$\diamondsuit$ | 6.72$\diamondsuit$ | 6.83$\diamondsuit$ | 6.69$\diamondsuit$ | 6.71$\diamondsuit$ | 6.73$\diamondsuit$ | 6.75$\diamondsuit$ |
| MLP | TyDi QA | 6.90$\diamondsuit$ | 7.72$\spadesuit$ | 6.96$\diamondsuit$ | 7.67$\spadesuit$ | 7.75$\spadesuit$ | 8.03$\spadesuit$ | 7.78$\spadesuit$ | 8.05$\spadesuit$ |
| | Hotpot QA | 6.83$\diamondsuit$ | 7.49$\spadesuit$ | 6.87$\diamondsuit$ | 7.52$\spadesuit$ | 7.50$\spadesuit$ | 8.02$\spadesuit$ | 7.53$\spadesuit$ | 8.06$\spadesuit$ |
| | MuSiQue | 6.87$\diamondsuit$ | 7.12$\diamondsuit$ | 6.91$\diamondsuit$ | 7.18$\diamondsuit$ | 7.51$\spadesuit$ | 8.04$\spadesuit$ | 7.54$\spadesuit$ | 8.05$\spadesuit$ |

Table 3 reports significant Spearman $\rho$ values for overlap between top-10 attention/MLP layers ranked by JSD and semantic gain. This frequent co-occurrence means that both metrics track the same retrieval signal that improves next-token prediction. Intuitively, when a layer genuinely draws on a retrieved sentence $c_i$ to write answers, ablating $c_i$ (i) alters that layer's token distribution yielding high JSD and (ii) removes "helpful push" toward correct tokens lowering semantic gain. Layers that merely supply generic syntax or parametric knowledge may boost semantic gain without changing under ablation, so their JSD remains low; strong overall correlation shows that such cases do not dominate, which further verifies effectiveness of ARC-JSD. Moreover, ARC-JSD is practical: it requires only forward passes, avoiding cost and saturation issues of gradient-based saliency (Qi et al., 2024). Unlike KL (undefined with zero-probability bins) or logit-space $\ell_2$ distances (scale-dependent) (Ferrando et al., 2023), JSD is finite, symmetric, scale-free, and measured in interpretable bits.

## 7 CASE STUDIES OF LOCATED ATTENTION HEADS AND MLPs

**Visualisation of MLPs.** Based on semantic gains analysis from § 6.2, we visualise projection of middle-residual stream $\mathbf{x}_i^{\ell,\text{mid}}$ and post-residual stream $\mathbf{x}_i^{\ell,\text{post}}$ via Logit Lens to vocabulary space in Fig. 3 and Appendix M. In Fig. 3, Qwen2-1.5B-IT was given data from TyDi QA dev dataset with context about mosquitos introduction from Wikipedia and query "*How many wings does a mosquito have?*" as input, and it generates responses "*A mosquito has two wings.*" as output. Based on our ARC-JSD method, we successfully located top-relevant context sentence, i.e., "*Mosquitoes have a slender segmented body, a pair of wings, three pairs of long hair-like legs, feathery antennae, and elongated mouthparts*". When we compare heatmap between $\mathbf{x}_i^{\ell,\text{post}}$ and $\mathbf{x}_i^{\ell,\text{mid}}$ in Fig. 3 from Layer 20 to Layer 27 (See Appendix M for full heatmap), we find that probability of correct token is in-

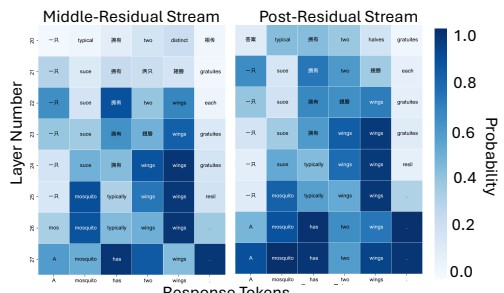

Figure 3: The projection of $\mathbf{x}_i^{\ell,\text{mid}}$ and $\mathbf{x}_i^{\ell,\text{post}}$ via Logit Lens to vocabulary space from layer 20 to layer 27 of Qwen2-1.5B IT in TyDi QA data sample, where the generated response $\mathcal{R}$ is "*A mosquito has two wings.*" (See Appendix M for all layer projections). Each cell shows the most probable token decoded via Logit Lens. The colour indicates the probability of the decoded token of the corresponding $\mathbf{x}_i^{\ell,\text{mid}}$ or $\mathbf{x}_i^{\ell,\text{post}}$.

creased significantly after $\mathbf{x}_i^{\ell,\text{post}}$ compared to $\mathbf{x}_i^{\ell,\text{mid}}$, such as 'wings' in Layer 23, 'A', 'has', 'two' in Layer 26, and 'mosquito', 'two', 'A' in Layer 27, which aligns with our findings that MLP contribute more parametric knowledge for context attribution in higher layers using JSD-based metric from § 5. Moreover, we find that several correct tokens are gradually transferred from their Chinese format to English version in Qwen2, such as '一只 (A)', '拥有 (has)' and '翅膀 (wings)', which is reasonable as Chinese is one of main language resources used in Qwen2 pre- and post-training (Yang et al., 2024). This finding also matches observations (Wu et al., 2025b) that representations tend to be anchored by semantically-equivalent dominant-language tokens in higher layers.

**Control RAGs based on ARC-JSD located Attention & MLPs.** We conduct an ablation study to compare JSD difference of responses by masking top-10 relevant attention heads and randomly-selected 10 attention heads in Table 5. Generally, ablating attention heads located by JSD metric causes larger JSD scores compared to random attention heads ablation, which further verifies the effectiveness of our proposed ARC-JSD method.

Table 4: Hallucination rate and factual information maintenance by gating top-5 JSD-based attention heads and MLP using Qwen2 7B IT on Hotpot QA dataset. †: Hallucination = RAG-generated response contradicts ground-truth while attributed context contained that ground-truth. ‡: we use JSD scores as a confidence gate to top-relevant attention heads and MLP layers with Mask $= 0.7 + 0.3 \times \text{sigmoid}(G)$, which means that top-relevant JSD attention and MLP will be multiplied with a mask close to 0.7 when their corresponding semantic gain $G$ is smaller than 0.

| Setup for Qwen2 7B IT | Hallucination Rate† | Pass@1 Factual F1 | Notes |
|---|---|---|---|
| Base RAG | 13.4% | 76.1 | One retrieved context sentence contains the answer |
| Gate Top-5 Attn & MLP‡ | 8.2% | 75.9 | Mask is 0.7 on top-5 JSD Attn & MLP when their $G < 0$ |
| Gate Random 5 Attn & MLP | 12.7% | 69.4 | Control |

To further evaluate how located internal components affect RAG behaviours, we conduct extra experiments on Qwen2 7B IT to show whether ARC-JSD can reduce hallucinations and maintain factual information unchanged. We select 200 data samples from Hotpot QA where Qwen2 7B IT did not provide a

Table 5: Comparison of average JSD scores between masking top-10 relevant attention heads and randomly masking 10 attention heads using all RAG models on all datasets.

| Top-10 Attention Heads | Random 10 Attention Heads |
|---|---|
| 2.23±0.12 | 1.53±0.76 |

truthful response compared to ground truth. From Table 4, we use JSD attribution scores and semantic gain $G$ as a confidence gate to reduce hallucination rate by $\approx 39\%$ without hurting factual F1 compared to random gating attention and MLPs[4].

**Causal Activation Ablation Study of ARC-JSD.**
We further conduct a causal activation ablation study by ablating each attention head and MLP layer for Qwen2-1.5B-IT on TyDi QA dataset. We mainly measure the percentage of context attribution accuracy decrease when each component is ablated. Fig. 4 demonstrates that those attention heads or MLP layers with higher JSD score lead to a larger context attribution accuracy drop, which aligns with findings of mechanistic study in Fig. 2.

**Token-level JSD distribution over Model's Response.** We conduct a case study to visualise token-level JSD distribution over the model's response *"A mosquito has two wings."*. Fig. 5 (a) clearly demonstrates that the token "two" contributes the highest JSD score change within the

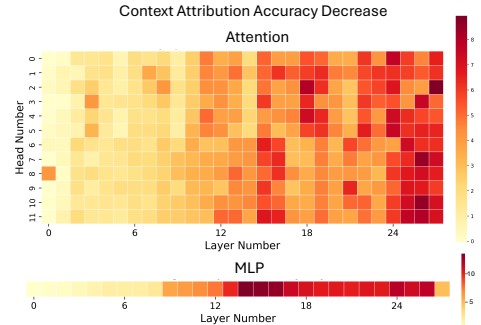

Figure 4: Causal activation ablation study of Qwen2-1.5B-IT on TyDi QA dataset for percentage of context attribution accuracy decrease by ablating each attention head and MLP layer.

whole response. For another example from TyDi QA with context about *"Burntisland Shipbuilding Company"*, the query is *"Who founded the Burntisland Shipbuilding Company?"*, and our ARC-JSD method successfully locates the top-related context sentence, i.e., *"Brothers Amos and Wilfrid Ayre founded Burntisland Shipbuilding Co. in 1918 as a First World War emergency shipyard. [1]"*. We further visualise model's response *"The Burntisland Shipbuilding Company was founded by brothers Amos and Wilfrid Ayre in 1918 during the First World War as an emergency shipyard."* in the Fig. 5 (b), and we can find that tokens "Am", "Wil", "Ay" and "re" contribute more JSD scores, which align with correct answer *"Amos and Wilfrid Ayre"*.

## 8 DISCUSSION

**Comparison JSD with KL, Wasserstein, TV and MMD.** In Fig. 6, we conduct an ablation study among different metrics, which shows that JSD works better than other metrics regarding compute-accuracy trade-off. Here, we discuss some possible reasons: *KL divergence* will explode whenever ablated run assigns $\approx 0$ probability to a token when full run uses (it is common in deep layers of LLMs). The unbounded scale makes it impossible to compare "how much layer 7 changed" to "how much layer 28 changed"; *TV* distance is bounded but too coarse, which means that two distributions that swap 5% mass on high-entropy tails give the same TV as two distributions that shift 5% mass off the top-1 token, yet the latter wrecks the answer; *Wasserstein* needs a distance between tokens. There is no canonical

---

[4]This experiment mainly evaluates how our ARC-JSD located attention and MLP affect the hallucinations and factual information in the RAG response based on the well-known agreement that attention heads and MLP focus on retrieval behavior and knowledge storage Sun et al. (2025), respectively.

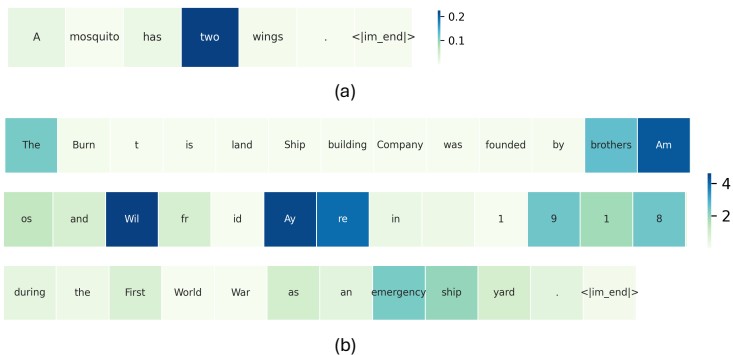

Figure 5: (a) Token-level JSD distribution over model's response *"A mosquito has two wings."*, where token "two" contributes the highest JSD score. (b) another example with a longer response, where tokens "Am", "Wil", "Ay" and "re" contribute more JSD scores.

ground metric on a 152K vocabulary (Qwen2-7B-Instruct version), and any choice (e.g., edit distance, embedding cosine, etc.) injects an orthogonal modelling assumption and costs $O(V^3)$ per layer; *MMD* always requires a kernel and a feature map to measure a Reproducing kernel Hilbert space (RKHS) norm, which is not tied to likelihood or entropy. It also needs a notion of distance between tokens to build the kernel (See Appendix L for details and examples).

**What if all JSD scores are very small?** When all scores are very small, it is the attribution. Small everywhere is not an error, and it means that RAG answers from parametric memory or retrieved passages are irrelevant. In those cases, we prefer to return "no evidence passage was used" rather than force-label the least-bad one. Practically, we can flag the answer with "low-evidence" when all sentence-JSD $< 0.02$ bits ($\approx$ median noise). The benefit than a threshold is that we can distinguish "no context used" from "weak but present context" without having to guess a universal cut-off. We could use that signal to re-query or warn the user, which is in practice a more faithful and safer behaviour than picking the least-small score.

**Generalisation of ARC-JSD to diverse and larger LLMs.** We evaluate our ARC-JSD using diverse and larger LLMs, i.e., LLaMA-3.1-8B-IT and Qwen3-Next-80B-A3B-IT, for generalisation evaluation. Fig. 7 shows that our ARC-JSD still has a better compute-accuracy trade-off compared to other baselines, which further demonstrates the effectiveness of ARC-JSD to diverse and larger LLMs based on Fig. 7 and Fig. 2.

**Comparison between single-hop QA and multi-hop QA under ARC-JSD.** We further take an analysis by comparing the single-hop QA example from the TyDi QA dataset and the multiple-hop QA examples from the Hotpot QA dataset and MuSiQue dataset in Appendix I.

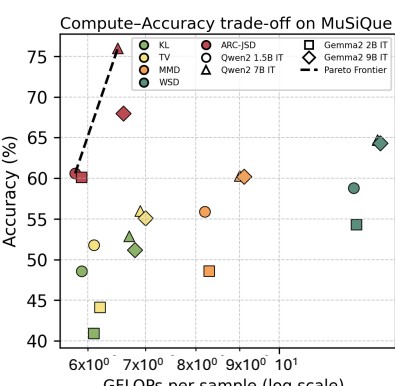

Figure 6: The compute-accuracy trade-off on MuSiQue for different metrics and ARC-JSD on 4 LLM backbones with GFLOPs $\log_{10}$ scale per sample.

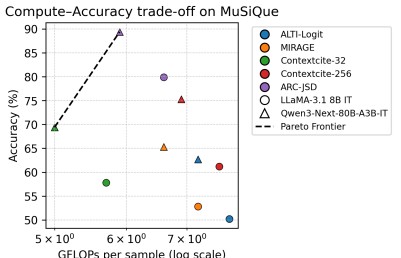

Figure 7: The compute-accuracy trade-off on MuSiQue for LLaMA-3.1-8B-IT and Qwen3-Next-80B-A3B-IT with GFLOPs $\log_{10}$ scale per sample.

We can find that ARC-JSD can directly attribute to the most relevant context that leads to the final answer with the highest JSD score. In addition, the sorted JSD ranking list further indicates how our ARC-JSD method takes causal reasoning to approach the final answer when we list the remaining top relevant contexts with their JSD scores. It brings a possibility that our ARC-JSD can be further used to help track the whole model's reasoning path for multi-hop QA settings, where we also provide more examples on medical and legal domains with higher demands for interpretability and attribution in Appendix J.

ACKNOWLEDGEMENT

This work is supported by the Gemma 2 Academic Program GCP Credit Award from Google and Google Cloud.

ETHICS STATEMENT

The proposed ARC-JSD uses open-source LLMs publicly available on HuggingFace, and all datasets are public question-answering domains and are publicly available, which did not raise any ethical concerns.

REPRODUCIBILITY STATEMENT

The open-source code of the proposed ARC-JSD is accessible from the anonymous repository: https://github.com/ruizheliUOA/ARC_JSD.

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

## A   APPENDIX

## B   BROAD IMPACT

RAG systems underpin a wide range of everyday activities, from itinerary planning and news aggregation to document drafting, by combining LLMs reasoning with evidence retrieved from external sources. Yet, the practical value of these systems hinges on our ability to verify that each generated statement is genuinely grounded in the retrieved material. The proposed *post-hoc* ARC-JSD method offers a lightweight, modular solution to this problem. Because ARC-JSD can be seamlessly integrated into any open-source RAG pipeline, it provides developers and researchers with an immediate way of auditing attribution fidelity, thereby strengthening the transparency, reliability, and ultimately the public trust in RAG-based applications.

## C   LIMITATIONS

Our work focuses on the analysis to (i) identify the context sentences that most strongly influence a RAG model's output and (ii) attribute that influence to specific attention heads and MLP layers via a JSD-based metric. Two important directions, therefore, remain unexplored. First, our layer-level view does not reveal which individual neurons within the MLPs mediate context attribution; techniques such as sparse autoencoder (SAE) probing could provide the necessary resolution. Second, we have not yet examined whether our proposed ARC-JSD method as a tool can be generalised to other tasks, such as membership inference attack. This is an exciting direction if the JSD attribution score can be used to locate which data is seen during training.

## D   DETAILS OF THE INTERNAL MECHANISMS OF LLMS

We consider the standard *autoregressive Transformer* architecture used in LLMs, originally introduced by Vaswani et al. (2017) and subsequently analysed in a series of mechanistic studies (Geva et al., 2021; Elhage et al., 2021; Geva et al., 2022; Dai et al., 2022; Meng et al., 2022a;b; Yuksekgonul et al., 2024). Given a prompt of length $T$, the input tokens $(t_1, \ldots, t_T)$ from the context-query pair, each drawn from a vocabulary $\mathcal{V}$, are mapped to $d$-dimensional embedding vectors $\mathbf{x}_i^0 \in \mathbb{R}^d$, where the embedding matrix $W_E \in \mathbb{R}^{|\mathcal{V}| \times d}$.

LLMs normally comprise $L$ identical layers. At layer $\ell$, the residual stream $\mathbf{X}^\ell = (\mathbf{x}_1^\ell, \ldots, \mathbf{x}_T^\ell)$, $\mathbf{x}_i^\ell \in \mathbb{R}^d$, acts as a common read–write buffer for both the multi–head attention and the MLP block (Elhage et al., 2021). For each token $i$, the residual update is

$$\mathbf{x}_i^\ell = \mathbf{x}_i^{\ell-1} + \mathbf{a}_i^\ell + \mathbf{m}_i^\ell, \tag{8}$$

where $\mathbf{a}_i^\ell$ and $\mathbf{m}_i^\ell$ denote the contributions of the attention and MLP sub-modules, respectively.[5]

---

[5]Layer normalisation preceding each sub-module is omitted here for clarity.

After the final layer, a LayerNorm $\sigma(\cdot)$ and the unembedding matrix $W_U \in \mathbb{R}^{d \times |\mathcal{V}|}$ produce the next-token distribution

$$\mathcal{P}_{\text{LM}}(t_{T+1} \mid t_{1:T}) = \text{softmax}(W_U \, \sigma(\mathbf{x}_T^L)). \tag{9}$$

Each layer contains $H$ attention heads, each factorised into $QK$ and $OV$ circuits operating with weight matrices $W_Q^{\ell,h}$, $W_K^{\ell,h}$, $W_V^{\ell,h}$, $W_O^{\ell,h} \in \mathbb{R}^{d \times d}$. The $QK$ circuit establishes the attention pattern $A^{\ell,h} \in \mathbb{R}^{T \times T}$, while the $OV$ circuit transports content across sequence positions. For head $h$ the contribution of source token $j$ to target token $i$ is

$$\mathbf{a}_{i,j}^{\ell,h} = A_{i,j}^{\ell,h} \left(\mathbf{x}_j^{\ell-1} W_V^{\ell,h}\right) W_O^{\ell,h}, \tag{10}$$

and the total attention update for token $i$ is

$$\mathbf{a}_i^\ell = \sum_{h=1}^{H} \sum_{j=1}^{T} \mathbf{a}_{i,j}^{\ell,h}. \tag{3}$$

A concise per-head summary is $\mathbf{a}_i^{\ell,h} = \sum_j \mathbf{a}_{i,j}^{\ell,h}$.

Following the key–value interpretation of MLP layers (Geva et al., 2021; Elhage et al., 2021), let $W_{\text{in}}^\ell \in \mathbb{R}^{d_{\text{m}} \times d}$ and $W_{\text{out}}^\ell \in \mathbb{R}^{d \times d_{\text{m}}}$ denote the input and output weights. Given $\mathbf{x}_i^{\ell-1}$, the block first produces coefficients

$$\mathbf{k}_i^\ell = \gamma(W_{\text{in}}^\ell \mathbf{x}_i^{\ell-1}) \in \mathbb{R}^{d_{\text{m}}}, \tag{11}$$

where $\gamma$ is the activation function (e.g. GELU). These coefficients weight the value vectors (rows of $W_{\text{out}}^\ell$) to yield

$$\mathbf{m}_i^\ell = \sum_{n=1}^{d_{\text{m}}} \mathbf{k}_i^{\ell,n} \mathbf{v}^{\ell,n}, \quad \mathbf{v}^{\ell,n} \equiv W_{\text{out}}^\ell[n,:]. \tag{12}$$

## E  EXPERIMENTAL DETAILS

We run all experiments using H100 GPUs, and we use the sentence tokeniser from the *nltk* library Bird et al. (2009) to preprocess all datasets. For all RAG models, i.e., Qwen2-1.5B-Instruct, Qwen2-7B-Instruct Yang et al. (2024), Gemma2-2B-Instruct and Gemma2-9B-Instruct Team et al. (2024), we use their standard chat templates to construct the prompt, i.e., using the context and query as a user's message.

When constructing prompts for TyDi QA dataset, we follow the prompt:

```
Context: {context}

Query: {question}
```

For Hotpot QA and MuSiQue datasets which have multiple documents for each data sample, the prompt is constructed as:

```
Title: {title_1}
Content: {document_1}
...
Title: {title_n}
Content: {document_n}

Query: {question}
```

## F    GPT-4.1 AS JUDGE FOR COMPARISON BETWEEN GENERATED RESPONSES OF RAG MODELS AND GOLD ANSWERS FROM DATASETS

After using our ARC-JSD to correctly locate the top relevant context sentences for generated responses, we further utilise GPT4.1 as a judge to check whether those responses correctly answer queries based on the corresponding context. As Table 6 shows, generated responses from all RAG models achieve high accuracy in successfully answering the queries based on the contexts, which demonstrates the fundamental ability of those instructed RAG models.

Table 6: GPT4.1 as a judge to evaluate the semantic equivalence between generated responses of RAG models and the corresponding gold answers from those datasets.

| Acc. (%) | Qwen2-1.5B-IT | Qwen2-7B-IT | Gemma2-2B-IT | Gemma2-9B-IT |
|---|---|---|---|---|
| TyDi QA | 99.1 | 99.4 | 98.9 | 99.5 |
| Hotpot QA | 99.2 | 99.5 | 99.1 | 99.6 |
| MuSiQue | 99.3 | 99.4 | 99.2 | 99.8 |

## G    THE USE OF LARGE LANGUAGE MODELS (LLMS) STATEMENT

This work was finished and written without the help of LLMs.

## H    COMPUTE-ACCURACY TRADE-OFF BETWEEN DIFFERENT BASELINES AND OUR ARC-JSD

We mainly compare the compute-accuracy trade-off between different baselines and our proposed ARC-JSD when attributing responses to relevant context. As Figure 8 and 9 show, our ARC-JSD method can achieve up to 3-fold speedup compared to other baselines. In addition, ARC-JSD is consistently Pareto-optimal over different LLM backbone sizes.

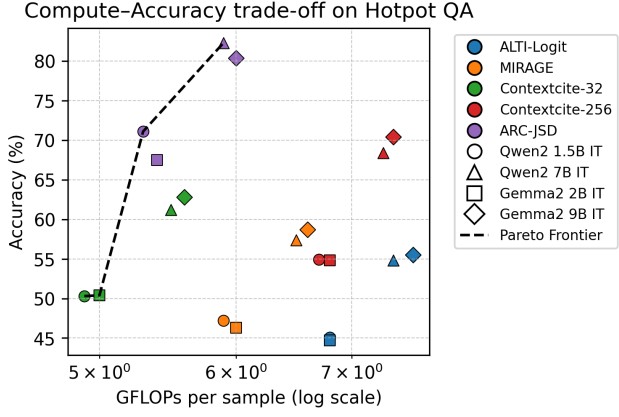

Figure 8: The compute-accuracy trade-off on Hotpot QA dataset for 4 baselines and ARC-JSD on 4 LLM backbones

## I    EXAMPLES OF ARC-JSD CONTEXT ATTRIBUTION

We demonstrate more examples of our ARC-JSD attribution method used for different RAG models on different datasets, where each example includes the query, generated responses and located top-1 sentence from the context.

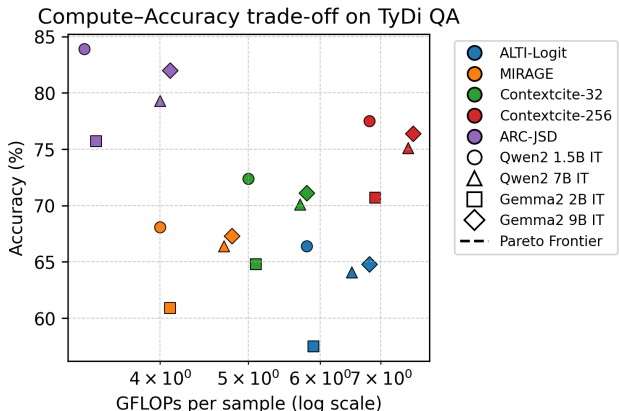

Figure 9: The compute-accuracy trade-off on TyDi QA dataset for 4 baselines and ARC-JSD on 4 LLM backbones

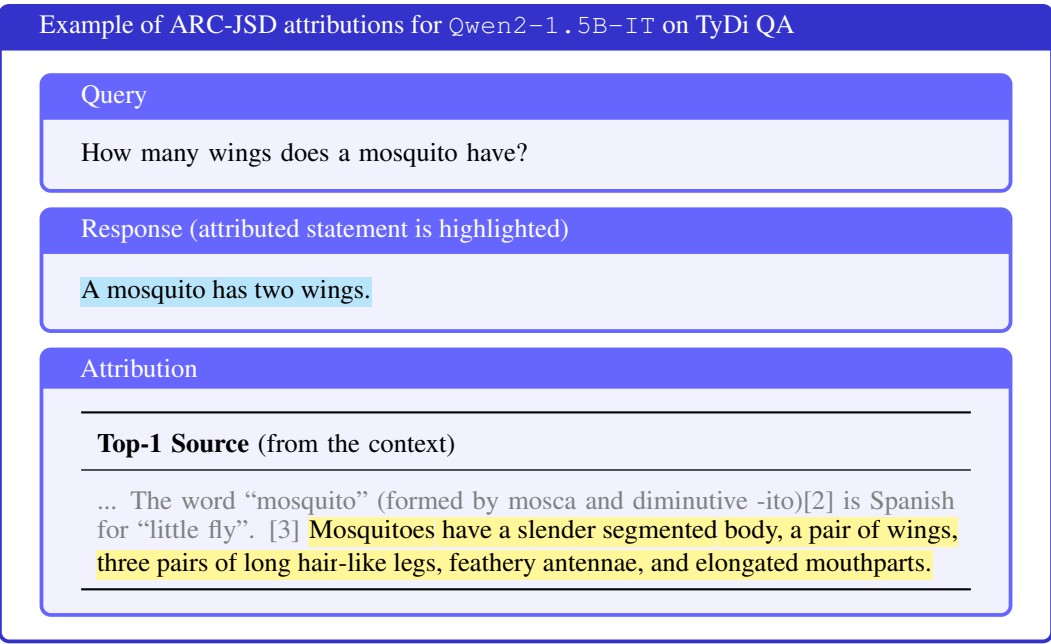

**Example of ARC-JSD attributions for `Qwen2-7B-IT` on MuSiQue**

**Query**

What is the date of birth of the person who was part of Ratata?

**Response (attributed statement is highlighted)**

The date of birth for Mauro Scocco, who was part of the pop group Ratata, is given as September 11, 1962.

**Gold Answer**

11 September 1962

**Attribution**

**Top-1 Source** (JSD: 18.64)

... Mauro Scocco (born 11 September 1962) is a Swedish pop artist of Italian descent. ...

**Top-2 Source** (JSD: 0.24)

... Title: Ratata (band) ...

**Top-3 Source** (JSD: 0.11)

... Ratata was a Swedish pop group fronted by Swedish vocalist Mauro Scocco formed in 1980. ...

**Top-4 Source** (JSD: 0.10)

... Title: Mauro Scocco ...

**Top-5 Source** (JSD: 0.03)

... After Ratata, Scocco has continued as a solo artist since. ...

**Top-6 Source** (JSD: 0.02)

... He has been described as "one of the sharpest songwriters in Sweden". ...

---

**Example of ARC-JSD attributions for** `Gemma2-2B-IT` **on TyDi QA**

**Query**

Who founded the Burntisland Shipbuilding Company?

**Response (attributed statement is highlighted)**

Amos and Wilfrid Ayre.

**Gold Answer**

Amos and Wilfrid Ayre

**Attribution**

**Top-1 Source** (JSD: 41.86)

... Brothers Amos and Wilfrid Ayre founded Burntisland Shipbuilding Co. in 1918 as a First World War emergency shipyard. ...

**Top-2 Source** (JSD: 0.12)

... Its yard at Burntisland West Dock had four berths and capacity to build ships up to 450 feet (140m) long[1] and up to 59 feet (18m) beam. ...

---

---

**Example of ARC-JSD attributions for** `Gemma2-9B-IT` **on Hotpot QA**

**Query**

Tom Thumb Food & Pharmacy is part of the southern division of a grocery company based in what city?

**Response (attributed statement is highlighted)**

The answer is **Boise, Idaho**. The text states that Tom Thumb Food & Pharmacy is part of the Southern division of Albertsons, and Albertsons Companies LLC is based in Boise, Idaho.

**Gold Answer**

Boise

**Attribution**

---

**Top-1 Source** (JSD: 6.73)

... It is also known in Brazil as "Chapolin", "Vermelhinho" ("Little Red One") and "Polegar Vermelho" ("Red Thumb") in allusion to the famous fairy tale character Tom Thumb. Title: Albertsons Albertsons Companies LLC is an American grocery company founded and based in Boise, Idaho. It is privately owned and operated by investors, including Cerberus Capital Management.

---

**Top-2 Source** (JSD: 6.14)

... It operate under the names Tom Thumb traditional grocery stores; Flagship Tom Thumb high end stores, usually in affluent areas. It makes up part of the Southern division of Albertsons. It is (as of May 2015) the number two supermarket in the competitive...

---

## J  ARC-JSD USED ON MEDICAL AND LEGAL TASKS

We further test our method by randomly selecting examples from the medical or legal datasets, e.g., PubMedQA, MedQuAD, and LegalBench, using the general instruction-tuning LLM backbones.

Based on those randomly selected examples below, we can find that our ARC-JSD is still applicable to the medical and legal domains. The top-1 located context with the highest JSD score correctly supports model's response. In addition, we can further understand how model's thinking about other context segments by looking at their JSD scores in the ranking list.

---

**Medical Example 1: Disease Symptoms and Characteristics**

**Context**

Parkinson's disease (PD) is a progressive neurodegenerative disorder that primarily affects movement. The hallmark motor symptoms include bradykinesia (slowness of movement), resting tremor, and rigidity. In addition to motor symptoms, patients often experience a wide range of non-motor symptoms, such as cognitive impairment, sleep disturbances, and autonomic dysfunction. The underlying pathology involves the loss of dopaminergic neurons in the substantia nigra, a region of the midbrain.

**Query**

What are the primary motor symptoms associated with Parkinson's disease?

**Response (attributed statement is highlighted)**

The primary motor symptoms associated with Parkinson's disease (PD) include:

1. Bradykinesia - Slowness of movement.
2. Resting tremor - A persistent shaking or trembling sensation during rest.
3. Rigidity - Difficulty moving certain muscles, especially when lying down or sleeping.

These symptoms can significantly impact daily activities and quality of life for individuals with PD.

**Attribution**

---

**Top-1 Source** (JSD: 4.89)

... The hallmark motor symptoms include bradykinesia (slowness of movement), resting tremor, and rigidity. ...

---

**Top-2 Source** (JSD: 0.63)

... Parkinson's disease (PD) is a progressive neurodegenerative disorder that primarily affects movement. ...

---

**Top-3 Source** (JSD: 0.15)

... In addition to motor symptoms, patients often experience a wide range of non-motor symptoms, such as cognitive impairment, sleep disturbances, and autonomic dysfunction. ...

---

**Top-4 Source** (JSD: 0.11)

... The underlying pathology involves the loss of dopaminergic neurons in the substantia nigra, a region of the midbrain. ...

---

**Medical Example 2: Adverse Effects and Contraindications**

**Context**

ACE inhibitors (e.g., lisinopril, ramipril) are commonly prescribed for hypertension and heart failure. A well-known side effect is a persistent, dry cough, which is thought to be mediated by bradykinin accumulation. More serious, though less common, adverse effects include hyperkalemia (high potassium levels) and angioedema (rapid swelling). These drugs are strictly contraindicated in pregnancy due to their teratogenic effects, particularly during the second and third trimesters.

**Query**

Why should ACE inhibitors not be used during pregnancy?

**Response (attributed statement is highlighted)**

ACE inhibitors, such as lisinopril and ramipril, are commonly prescribed for hypertension and heart failure. However, they have been found to cause several serious adverse effects when taken during pregnancy, including hyperkalemia (high potassium levels) and angioedema (rapid swelling). Additionally, these drugs are strictly contraindicated in pregnancy due to their teratogenic effects, particularly during the second and third trimesters. Therefore, it is important to avoid using ACE inhibitors during pregnancy to minimize the risk of harm to both mother and baby.

**Attribution**

**Top-1 Source** (JSD: 20.31)

... These drugs are strictly contraindicated in pregnancy due to their teratogenic effects, particularly during the second and third trimesters. ...

**Top-2 Source** (JSD: 16.71)

... More serious, though less common, adverse effects include hyperkalemia (high potassium levels) and angioedema (rapid swelling). ...

**Top-3 Source** (JSD: 9.83)

... ACE inhibitors (e.g., lisinopril, ramipril) are commonly prescribed for hypertension and heart failure. ...

**Top-4 Source** (JSD: 1.05)

... A well-known side effect is a persistent, dry cough, which is thought to be mediated by bradykinin accumulation. ...

**Legal Example 1: Criminal Procedure (Miranda Rights)**

**Context**

The Miranda v. Arizona ruling by the Supreme Court established that custodial interrogation is inherently coercive. To safeguard the Fifth Amendment privilege against self-incrimination, law enforcement officials must inform a suspect in custody of their rights before questioning begins. These rights include the right to remain silent, the warning that anything said can be used against them in court, the right to an attorney, and the right to have an attorney appointed if they cannot afford one. A waiver of these rights must be made voluntarily, knowingly, and intelligently.

**Query**

What specific rights must a suspect in custody be informed of before being questioned, according to the Miranda ruling?

**Response (attributed statement is highlighted)**

According to the Miranda ruling, a suspect in custody must be informed of their rights before being questioned, including the following:

1. The right to remain silent.
2. The warning that anything said can be used against them in court.
3. The right to an attorney.
4. The right to have an attorney appointed if they cannot afford one.

These rights must be informed voluntarily, knowingly, and intelligently.

**Attribution**

**Top-1 Source** (JSD: 8.65)

... These rights include the right to remain silent, the warning that anything said can be used against them in court, the right to an attorney, and the right to have an attorney appointed if they cannot afford one. ...

**Top-2 Source** (JSD: 6.10)

... A waiver of these rights must be made voluntarily, knowingly, and intelligently. ...

**Top-3 Source** (JSD: 0.73)

... To safeguard the Fifth Amendment privilege against self-incrimination, law enforcement officials must inform a suspect in custody of their rights before questioning begins. ...

**Top-4 Source** (JSD: 0.54)

... The Miranda v. Arizona ruling by the Supreme Court established that custodial interrogation is inherently coercive. ...

**Legal Example 2: Intellectual Property (Copyright Fair Use)**

**Context**

Section 107 of the U.S. Copyright Act provides for the "fair use" of a copyrighted work without the owner's permission for purposes such as criticism, comment, news reporting, teaching, scholarship, or research. To determine if a particular use is "fair," courts evaluate four statutory factors. These factors are: (1) the purpose and character of the use, including whether it is commercial or for non-profit educational purposes; (2) the nature of the copyrighted work; (3) the amount and substantiality of the portion used; and (4) the effect of the use upon the potential market for the work.

**Query**

What are the four factors a court uses to determine if the use of copyrighted material qualifies as 'fair use'?

**Response (attributed statement is highlighted)**

The four factors that a court uses to determine if the use of copyrighted material qualifies as 'fair use' are:

1. The purpose and character of the use, including whether it is commercial or for non-profit educational purposes.
2. The nature of the copyrighted work.
3. The amount and substantiality of the portion used.
4. The effect of the use upon the potential market for the work.

These factors help courts assess whether the use of copyrighted material falls within the bounds of fair use and does not infringe on the copyright holder's rights.

**Attribution**

**Top-1 Source** (JSD: 21.45)

... These factors are: (1) the purpose and character of the use, including whether it is commercial or for non-profit educational purposes; (2) the nature of the copyrighted work; (3) the amount and substantiality of the portion used; and (4) the effect of the use upon the potential market for the work. ...

**Top-2 Source** (JSD: 0.41)

... To determine if a particular use is "fair," courts evaluate four statutory factors. ...

**Top-3 Source** (JSD: 0.27)

... Section 107 of the U.S. Copyright Act provides for the "fair use" of a copyrighted work without the owner's permission for purposes such as criticism, comment, news reporting, teaching, scholarship, or research. ...

# K JSD-BASED MECHANISTIC INSIGHTS FOR LOCATED ATTENTION HEADS AND MLPS

We visualise more attention heads and MLP heatmaps using our JSD-based mechanistic approach, where we can find that most RAG models include attribution-relevant attention heads and MLPs across the intermediate and higher layers. On the Hotpot QA and MuSiQue datasets, Gemma2-2B-IT has some relevant attention heads on the lower layers.

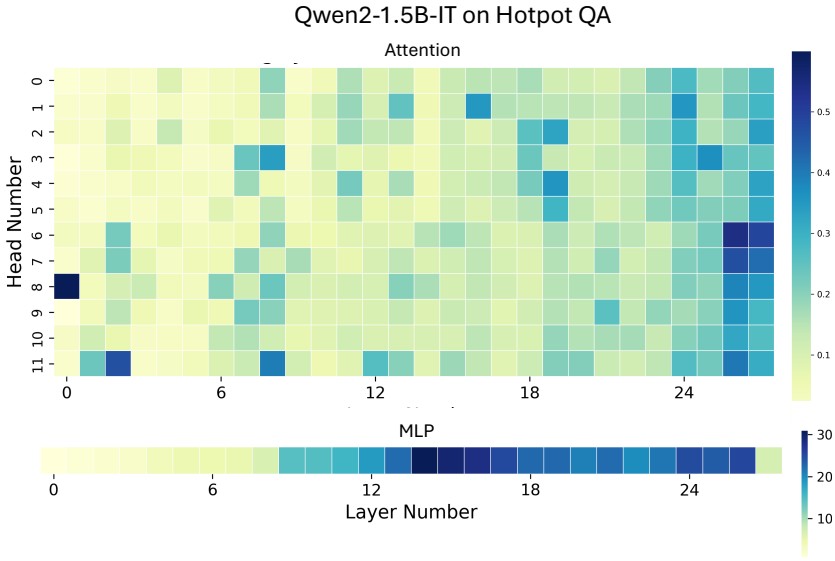

Figure 10: The average JSD score of attention heads and MLP of Qwen2-1.5B-IT on Hotpot QA dataset across all layers. The deeper colour indicates larger JSD scores.

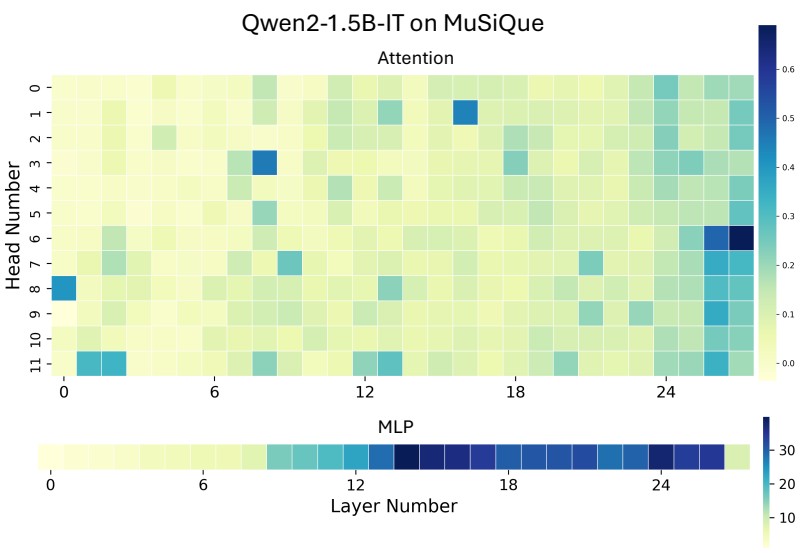

Figure 11: The average JSD score of attention heads and MLP of Qwen2-1.5B-IT on MuSiQue dataset across all layers. The deeper colour indicates larger JSD scores.

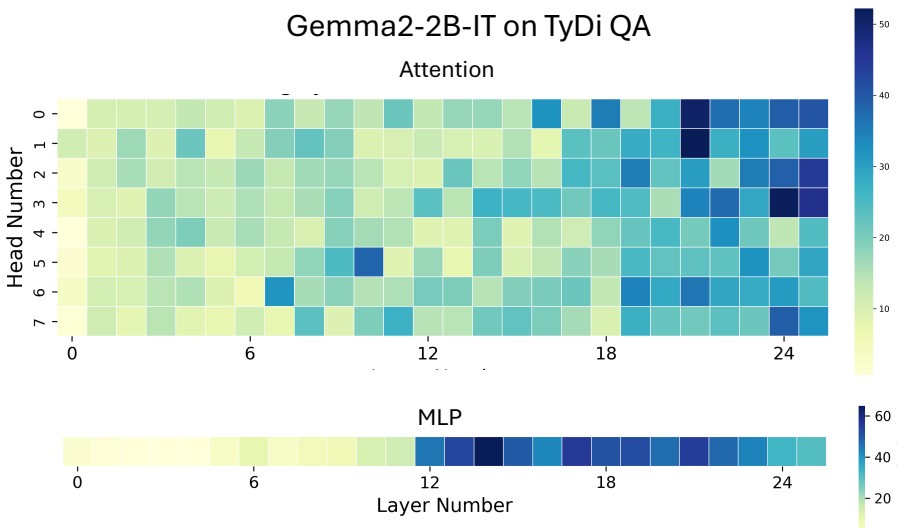

Figure 12: The average JSD score of attention heads and MLP of Gemma2-2B-IT on TyDi QA dataset across all layers. The deeper colour indicates larger JSD scores.

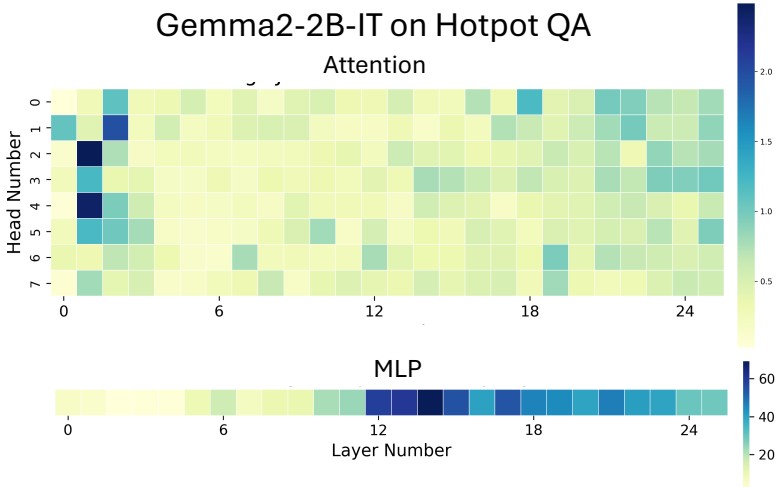

Figure 13: The average JSD score of attention heads and MLP of Gemma2-2B-IT on Hotpot QA dataset across all layers. The deeper colour indicates larger JSD scores.

## L  COMPARISONS OF JSD WITH KL, WASSERSTEIN, TV AND MMD IN DETAIL

**Direct log-probability or KL Divergence.**   Most existing baselines, e.g., ContextCite Cohen-Wang et al. (2024), SelfCite Chuang et al. (2025) and AttriBoT Liu et al. (2024), use direct log-probability or KL divergence as metric for context attribution. However, these metrics drop diverges if the masked run assigns $\approx 0$ probability to the token, which is sensitive to highly-skewed token frequencies. Moreover, if JSD is replaced with KL in the Eq. 1 and Eq. 2, it will bring some influence to the attribution impact:

- **Asymmetry / direction choice**: We must choose $\mathrm{KL}(P|Q)$ or $\mathrm{KL}(Q|P)$. The ranking of sentences can flip depending on direction. There is no principled reason to prefer one for attribution. Using the symmetrized Jeffreys divergence $\mathrm{KL}(P|Q) + \mathrm{KL}(Q|P)$ removes directionality, but it does not

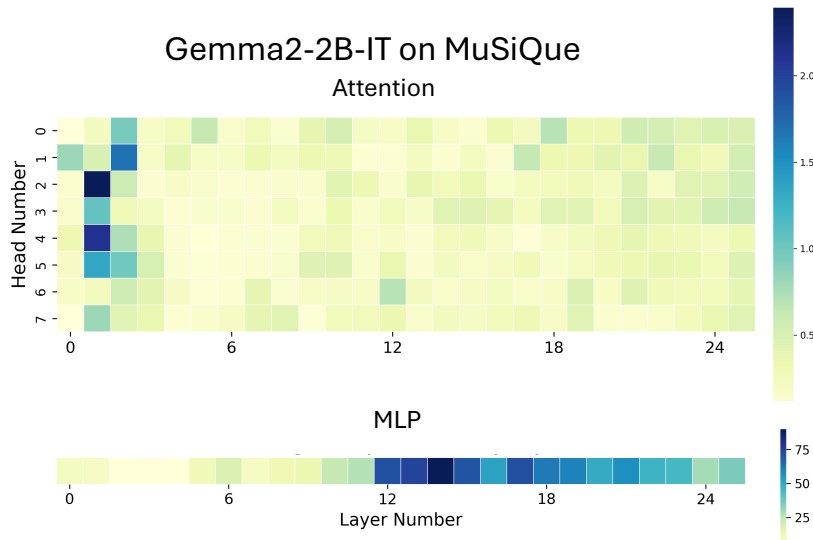

Figure 14: The average JSD score of attention heads and MLP of Gemma2-2B-IT on MuSiQue dataset across all layers. The deeper colour indicates larger JSD scores.

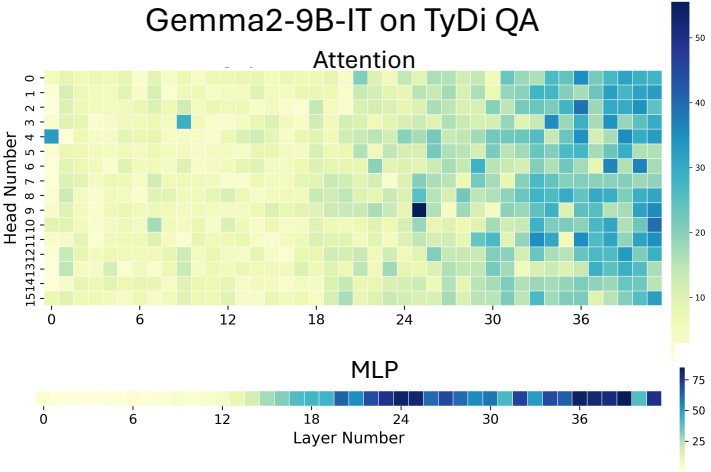

Figure 15: The average JSD score of attention heads and MLP of Gemma2-9B-IT on TyDi dataset across all layers. The deeper colour indicates larger JSD scores.

fix the core issues, such as unboundedness, tail sensitivity, numerical instability, and lack of a common scale.

- **Unbounded & numerically unstable**: If the ablated run puts (near) zero mass on a token that the full run assigns mass to (that is common at deeper layers), KL explodes or becomes extremely noisy unless we add ad-hoc smoothing. However, this tends to overweight tail events and can produce false positives.

- **Cross-layer incomparability**: Because KL or Jeffreys are unbounded, a few positions with tiny denominators dominate the sentence score, i.e., comparing "how much layer 7 changed" vs "layer 28" becomes unstable. JSD's boundedness is crucial for consistent ranking and aggregation.

Therefore, if we replace JSD with KL, there will be lower precision/recall for "relevant sentence" ranking (it brings more variance, dependence on $\varepsilon$ and direction), which will further lead to low

attribution accuracy. It will also tend to disagree more with independent, behaviour-aligned probes (e.g., semantic gain used in our work), although KL divergence has the same FLOPs as JSD.

**Wasserstein Distance.** Assume we choose a ground metric $c(a, b)$ over tokens (e.g., token-Hamming, character edit distance, or embedding-cosine cost), and use entropic-regularised Sinkhorn for Wasserstein. When we replace JSD with Wasserstein distance, it will affect attribution:

- **Metric choice drives the result**: Edit distance and embedding-cosine will encode orthography or static similarity, not decoding behaviour. They may call a move toward a typo-like token "cheap" and a move toward a semantically correct rival "expensive", which misaligns with which changes actually flip the output (See more detailed discussion below).

- **Hyperparameters matter**: Sinkhorn $\varepsilon$ (regularisation) and number of iterations change the scale and ranking. Different reasonable settings can reorder "relevant sentences".

- **Context dependence missing**: A single cost matrix $c(a, b)$ ignores that token meaning is position- and layer-dependent in a transformer-based LLM, which means that we either accept a mismatch or introduce layer-specific cost matrices (which becomes circular and heavy).

So, rankings become sensitive to modelling choices not tied to the LM's probability geometry, typically reducing correlation with behaviour (semantic gain) and causal precision in context attribution.

For FLOPs comparison, if we use full support for Wasserstein distance, Sinkhorn per pair costs $O(KV^2)$ operations (and $O(V^2)$ memory) for $K$ iterations, where $V \sim 152k$. Instead, if we use top-$k$ support trick, we restrict to top-$k$ tokens of $P$ and $Q$ (say $k \in [100, 500]$). Cost becomes $O(Kk^2)$ per $(layer, r_j)$, plus top-$k$ selection $O(V log k)$. This is still orders of magnitude above JSD in practice and adds hyperparameters $k, K, \varepsilon$.

**MMD Metric.** Let $k(\cdot, \cdot)$ be a kernel on tokens; for categorical distributions one computes $\text{MMD}^2(P, Q) = (P - Q)^\top K(P - Q)$ with $K_{ab} = k(a, b)$. When we replace JSD with MMD, it will affect attribution:

- **Kernel choice = modelling assumption**: We need to make multiple choices: Gaussian or Laplace on which embeddings? What bandwidth? Results (and rankings) will vary with these choices.

- **Units & interpretability**: Values depend on kernel scale and there is no direct link to entropy or cross-entropy (which govern decoding). The equal mass moves on tail tokens can dominate if the kernel puts them in "diverse" regions, even though they don't affect behaviour.

- **Edge case**: If we set $k(a, b) = \mathbf{1}[a=b]$, MMD reduces to $\ell_2$ on probabilities, again misaligned with decoding (uniformly weights all coordinates).

So, replacing JSD with MMD will bring more sensitivity to hyperparameters, and it has weaker correlation with behaviour, and less stable cross-layer comparisons than JSD.

For FLOPs comparison, if we use dense kernel, a naive computation is $O(V^2)$ per $(layer, r_j)$ (matrix–vector with $K \in \mathbb{R}^{V \times V}$). Instead, if we use Low-rank/Nyström rank $r$, it will cost $O(rV)$ per pair, but we must tune $r$ and store factors. With $r = 256$, this is $\sim 256\times$ the work of JSD's $O(V)$ reduction, and quality also depends on $r$. We also need to consider plus kernel selection/bandwidth tuning overhead.

**Using edit distance or embedding cosine as metrics for Wasserstein or MMD.** When Wasserstein or MMD uses edit distance or embedding cosine as metrics, it has several limitations:

1. Edit distance (token/character level):

    - **Tokenisation mismatch**: In subword vocabularies, a single semantic change can span many subwords, and edit distance on token strings becomes an artefact of the tokeniser, not semantics.

    - **Semantic blindness**: For the example: "Paris" $\rightarrow$ "Lyon" (same POS, both cities) and "Paris" $\rightarrow$ "Party". At the token level, any substitution has unit cost, so replacing "Paris" with either "Lyon" or "Party" is equally cheap, despite radically different semantic consequences. With subword tokenisation, the cost becomes tokeniser-dependent. Character-level edit distance

differentiates orthography (e.g., "Party" is closer to "Paris" than "Lyon"), which misaligns with factual attribution

- **Decoding irrelevance**: The decoder's choice is driven by probability mass, not string operations. A small edit distance can correspond to a huge shift in probability, and vice versa.

2. Embedding-cosine ground metrics:

- **Context dependence**: Token meaning in transformers is contextual. A static vocab-level embedding (or even the unembedding vectors) is not the representation used at the position/layer where attribution is measured. A faithful ground metric would need position- and layer-specific distances, which will explode in complexity and introduce circularity.
- **Anisotropy & polysemy**: Cosine distances in high-dimensional language embeddings are known to concentrate and to blur senses, which means that "nearby" vectors can still correspond to different factual claims. Wasserstein might then deem a large semantic change "cheap to move," underestimating its effect on generation.
- **Tunable choices**: Which embedding? Which layer? Do we normalise? Each choice changes the cost matrix and can alter the ranking of "relevant" layers and context sentences, which is exactly the orthogonal modelling assumption we seek to avoid.

**TV Metric.** Here, we provide a simple example to explain why TV distance is not an ideal metric to use for context attribution.

The definition of TV distance for two discrete distributions $P, Q$ over the same vocabulary is:

$$\text{TV}(P, Q) = \frac{1}{2} \sum_t |P(t) - Q(t)| \tag{13}$$

Here, TV measures the total amount of probability mass moved, but not where it moved.

For any decoding methods used in LLMs, they are more affected by the position where probability mass moved, e.g., greedy decoding picks the token with the largest probability, or sampling and beam search are also dominated by how mass is distributed among the top few tokens.

Here is one example to consider a single decoding step with three candidate tokens: $t_1$ = the ground-truth/desired token, $t_2$ = a strong competitor, $t_3$ = a low-probability tail token.

Let the full-context distribution at one decoding step be:

$$P = \big[p(t_1), p(t_2), p(t_3)\big] = [0.52, 0.43, 0.05] \tag{14}$$

Consider two different ablated distributions that both move the same amount of mass $\varepsilon = 0.05$:

*Case A— move mass in the tail (does not flip the output prediction)*:

Shift $\varepsilon$ from $t_3$ (tail) to $t_2$: i.e., $Q_{\text{tail}} = [0.52, 0.48, 0.00]$. TV calculation will be:

$$\text{TV}(P, Q_{\text{tail}}) = \tfrac{1}{2}\big(|0.52-0.52| + |0.43-0.48| + |0.05-0.00|\big) = \tfrac{1}{2}(0 + 0.05 + 0.05) = 0.05. \tag{15}$$

When we use greedy choice, we still choose $t_1$ because 0.52 remains the largest.

*Case B — move mass off the top onto its nearest competitor (does flip the output prediction)*:

Shift the same $\varepsilon = 0.05$ from $t_1$ to $t_2$: $Q_{\text{top}} = [0.47, 0.48, 0.05]$. TV calculation will be:

$$\text{TV}(P, Q_{\text{top}}) = \tfrac{1}{2}\big(|0.52-0.47| + |0.43-0.48| + |0.05-0.05|\big) = \tfrac{1}{2}(0.05 + 0.05 + 0) = 0.05. \tag{16}$$

When we use greedy choice, it will flip to $t_2$ because $0.48 > 0.47$.

Both perturbations have the same TV = 0.05, but only Case B changes the token the model outputs.

If we move $\varepsilon$ probability from any token i to any token j (and leave all others unchanged), the absolute differences are $|-\varepsilon|$ for i, $|+\varepsilon|$ for j, and 0 elsewhere, so $\text{TV}(P, Q) = \frac{1}{2}\big(\varepsilon + \varepsilon\big) = \varepsilon$, regardless of which tokens i and j you chose, Which means that TV "sees" only the amount moved, not where it came from or went.

Yet output behaviour depends critically on where the mass moves:

- The arg-max flips when $p_j + \varepsilon > p_i - \varepsilon \iff \varepsilon > \frac{1}{2}(p_i - p_j)$. In our numbers, $p_1 - p_2 = 0.09$, so any $\varepsilon > 0.045$ flips the token, where Case B does ($\varepsilon = 0.05$), Case A does not.

- For sampling, the log-odds change by $\Delta \log \frac{p_i}{p_j} = \log \frac{p_i - \varepsilon}{p_j + \varepsilon} - \log \frac{p_i}{p_j}$, which is large and negative only when you move mass between the top competitors (Case B), not when you shuffle tail mass (Case A). But TV assigns both moves the same distance.

In addition, we conduct ablation studies to compare different metrics with JSD on compute-accuracy trade-off on different datasets with 4 LLM backbones. As shown in Fig. 16 and Fig. 17, JSD yields a better compute-accuaracy trade-off than other metrics.

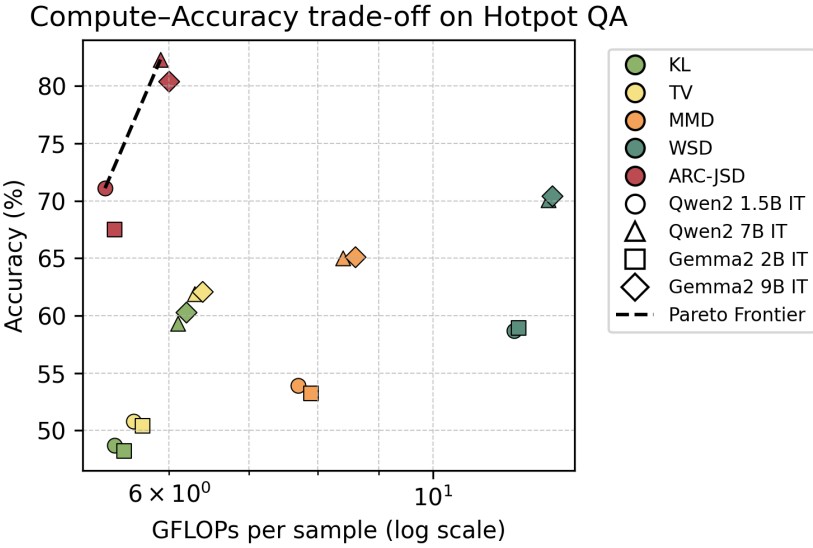

Figure 16: The compute-accuracy trade-off on Hotpot QA for different metrics and ARC-JSD on 4 LLM backbones with GFLOPs $\log_{10}$ scale per sample.

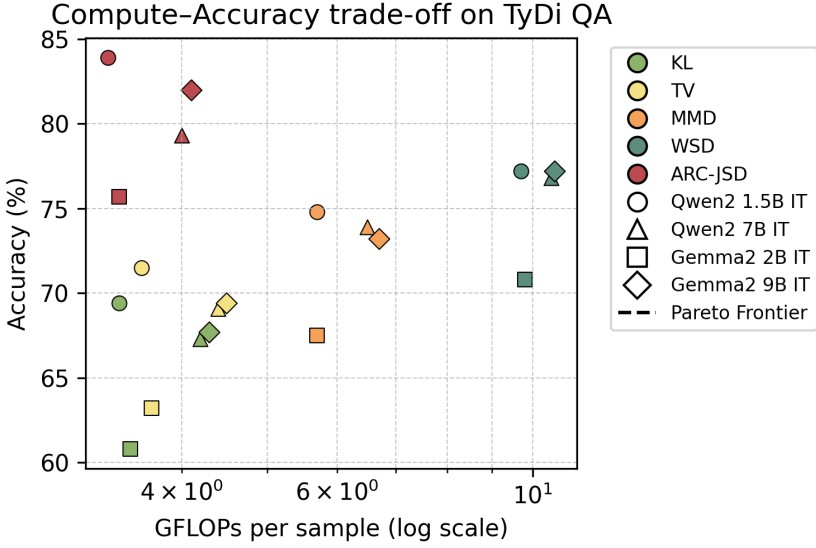

Figure 17: The compute-accuracy trade-off on TyDi QA for different metrics and ARC-JSD on 4 LLM backbones with GFLOPs $\log_{10}$ scale per sample.

# M  CASE STUDIES OF ATTENTION AND MLP'S CONTRIBUTION FOR EACH RESPONSE TOKEN

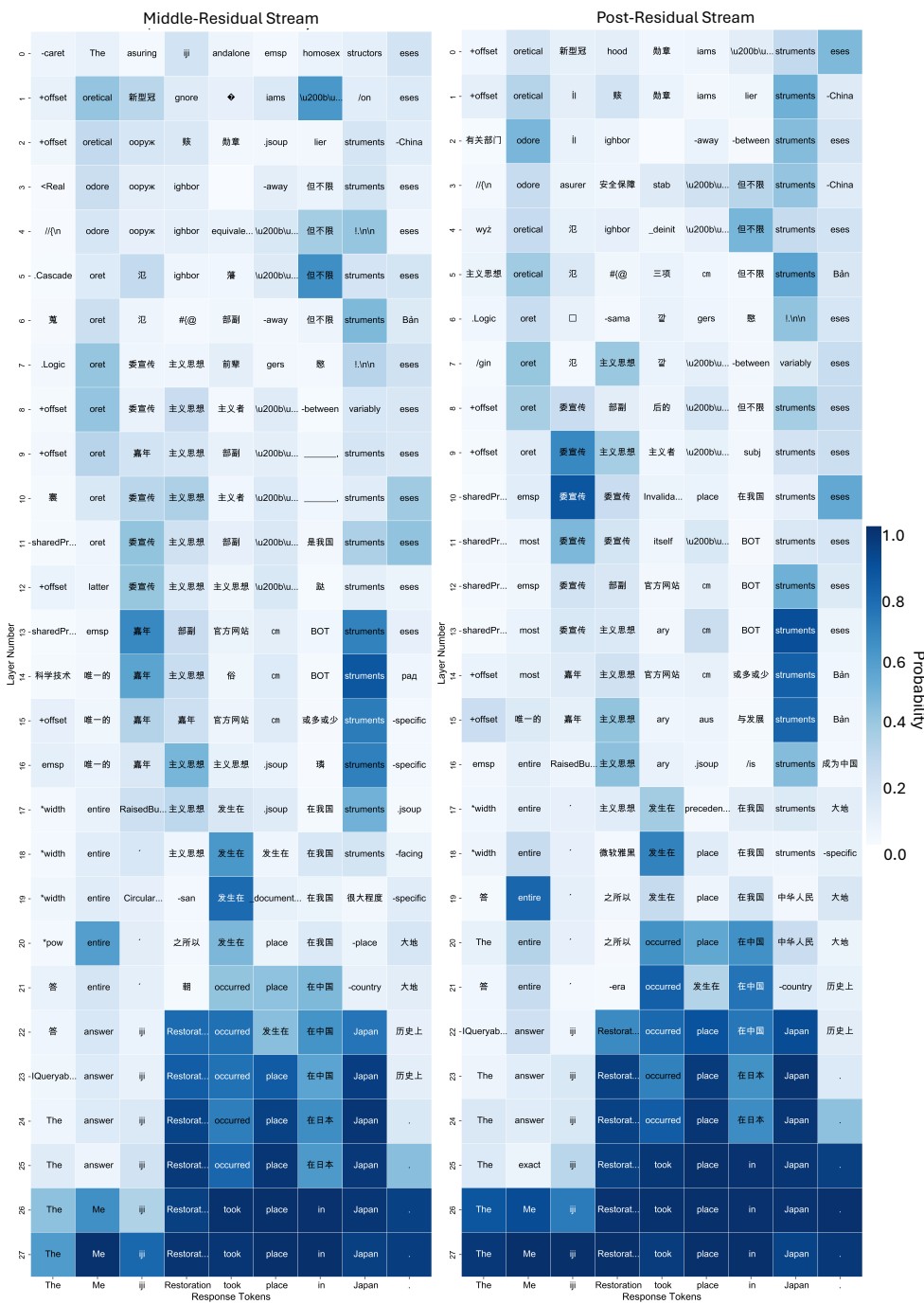

Figure 18: The projection of $\mathbf{x}_i^{\ell,\mathrm{mid}}$ and $\mathbf{x}_i^{\ell,\mathrm{post}}$ via Logit Lens to vocabulary space from layer 20 to layer 27 of Qwen2-1.5B IT in TyDi QA data sample, where the generated response $\mathcal{R}$ is "*The Meiji Restoration took place in Japan.*". Each cell shows the most probable token decoded via Logit Lens.

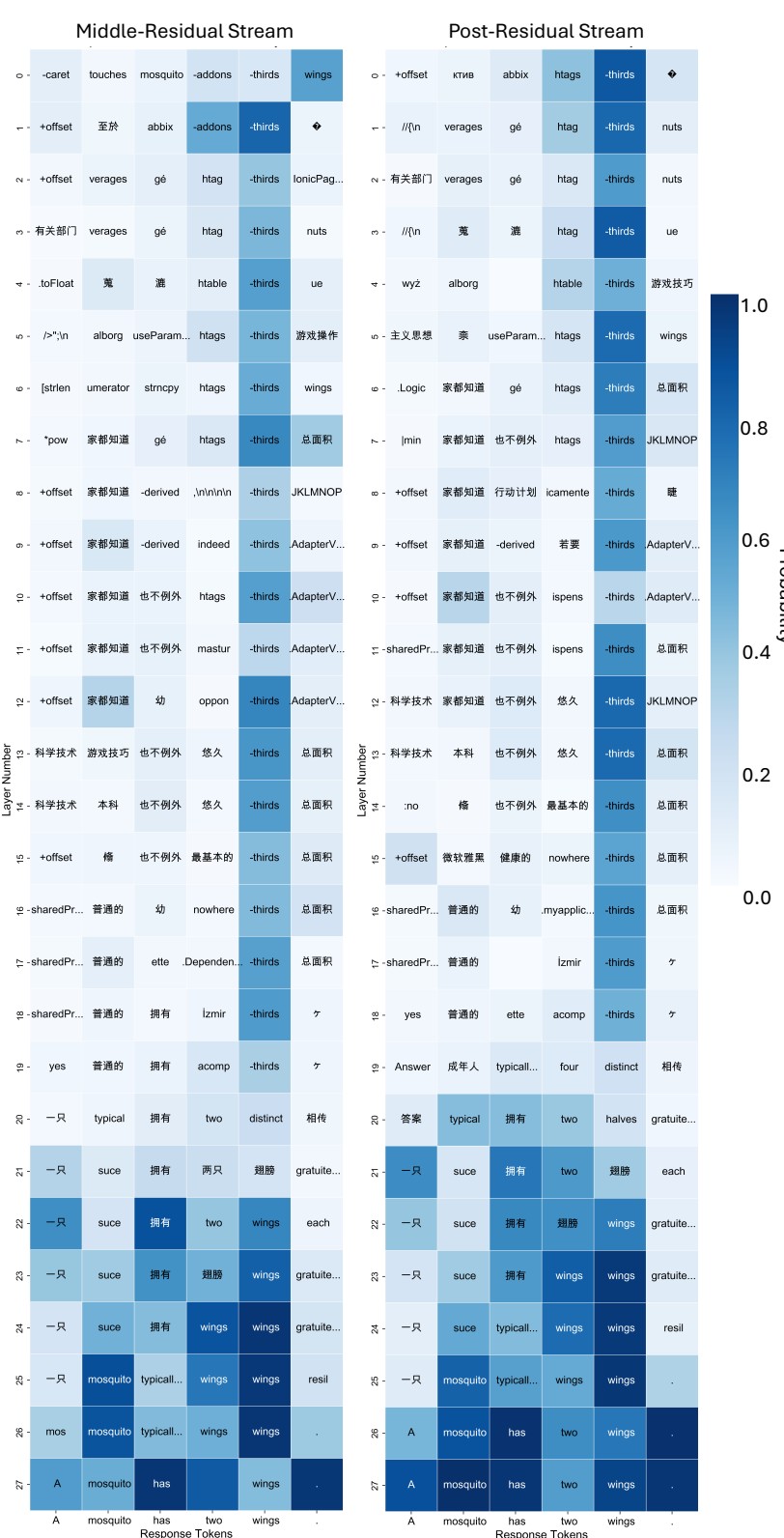

Figure 19: The projection of $\mathbf{x}_i^{\ell,\mathrm{mid}}$ and $\mathbf{x}_i^{\ell,\mathrm{post}}$ via Logit Lens to vocabulary space from layer 20 to layer 27 of Qwen2-1.5B IT in TyDi QA data sample, where the generated response $\mathcal{R}$ is "*A mosquito has two wings.*". Each cell shows the most probable token decoded via Logit Lens. The colour indicates the probability of the decoded token of the corresponding $\mathbf{x}_i^{\ell,\mathrm{mid}}$ or $\mathbf{x}_i^{\ell,\mathrm{post}}$.

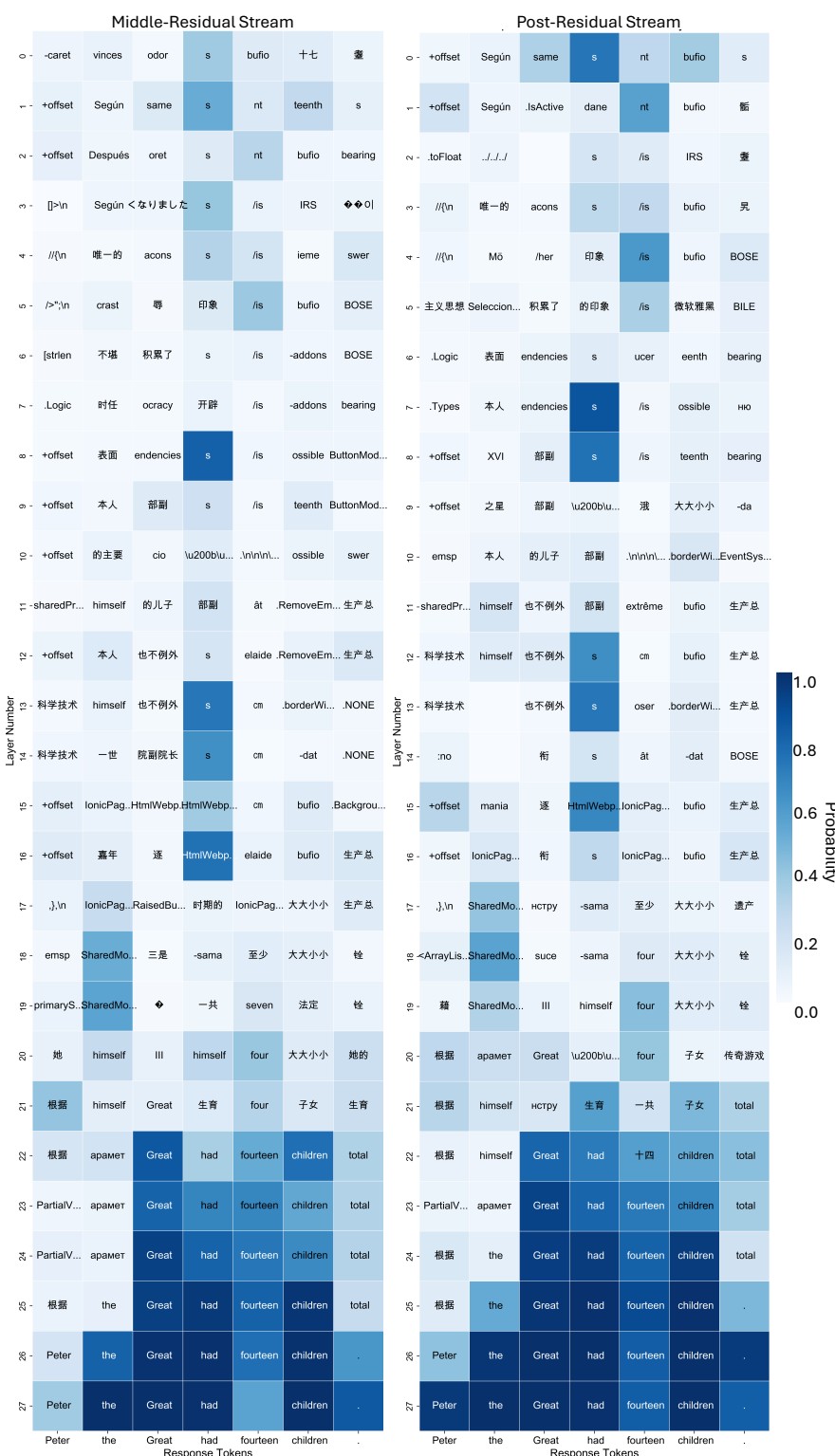

Figure 20: The projection of $\mathbf{x}_i^{\ell,\text{mid}}$ and $\mathbf{x}_i^{\ell,\text{post}}$ via Logit Lens to vocabulary space from layer 20 to layer 27 of Qwen2-7B IT in TyDi QA data sample, where the generated response $\mathcal{R}$ is "*Peter the Great had fourteen children.*". Each cell shows the most probable token decoded via Logit Lens.

