# OpenReview forum: "Attributing Response to Context: A Jensen–Shannon Divergence Driven Mechanistic Study of Context Attribution in Retrieval-Augmented Generation"
_ICLR.cc/2026/Conference — ICLR 2026 Poster_

### Official Review · Reviewer_BYoP · 2025-10-30

**Soundness:** 4
**Presentation:** 3
**Contribution:** 3
**Rating:** 6
**Confidence:** 2

**Summary:**

This paper proposes ARC-JSD for RAG, which quantifies the Jensen–Shannon Divergence between response distributions generated with full and sentence-ablated contexts. This divergence serves as a reliable signal for identifying which context sentences most influence model outputs. The authors demonstrate that ARC-JSD achieves around 10% higher attribution accuracy and up to threefold computational efficiency compared to existing baselines.

**Strengths:**

1.The writing is fluent and the proposed method is easy to understand.

2.The theoretical analysis is well-developed, effectively explaining why JSD can serve as a valid metric for attribution analysis.

3.The experimental evaluation is comprehensive, demonstrating both the efficiency and effectiveness of the proposed approach.

**Weaknesses:**

1.Lack of clear definition. The paper does not appear to provide a formalized definition of the entire response distribution; while the token-level distribution is intuitive, the implementation details for computing the distribution over the full generated response should be clarified.

2.Limited model diversity: Although the evaluation is comprehensive, the models used are primarily from the Qwen and Gemma families at relatively small scales. It is recommended to include other representative open-source architectures such as LLaMA, and to analyze larger-scale models (e.g., 70B parameters) for more generalizable insights.

**Questions:**

1.How does this approach address real-world complex queries where accurate responses typically require the integration of multiple context segments?

2.How is the granularity of context segmentation determined, and what impact does this granularity have on the attribution performance?

---

> ### Author Response · Authors · 2025-11-20
> **Reply by Authors**
>
> **W1**:
> > Lack of clear definition. The paper does not appear to provide a formalized definition of the entire response distribution; while the token-level distribution is intuitive, the implementation details for computing the distribution over the full generated response should be clarified.
>
> **A1**:
>
> We apologise for such an unclear definition. In Section 3, we give a definition about the entire response distribution conditioned on the full context and ablated context:
>
> $\mathcal{R} \sim \mathcal{P}_{\text{LM}}\(\cdot|c_1,\cdots,c_C,\mathcal{Q}\)$
>
> $\mathcal{R} \sim \mathcal{P}_{\text{LM}}\(\cdot|\mathcal{C}_A(c_i),\cdots,c_C,\mathcal{Q}\)$
>
> where $\mathcal{C}_{A}(c_i)=\mathcal{C}\setminus \\{c_i\\},i\in \\{1, \cdots, |\mathcal{C}|\\}$
>
> During the implementation of the computation about the distribution over the full generated response, we sum up the token-level JSD between each response token under the full context and ablated context conditions, which is the contribution of the whole response, i.e., equation 1 in our work.
>
> The corresponding code segment is below:
>
> ```python
>
> JSD_div_values = []
> for i, ablated_context in enumerate(ablated_contexts):
>     ablated_chat_prompt_ids, _ = self.get_prompt_id(ablated_context, input_query)
>     ablated_input_ids = torch.tensor([ablated_chat_prompt_ids + output_ids], device=self.model.device)
>     with torch.no_grad():
>         ablated_output = self.model(input_ids=ablated_input_ids)
>     ablated_logits = ablated_output.logits[:, -(len(output_ids) + 1): -1].bfloat16()
>
>
>     jsd = self.calculate_dist_2d(ablated_logits.squeeze(0), full_logits.squeeze(0))
>     JSD_div_values.append(jsd)
>
> def calculate_dist_2d(self, ablated_context_logits, full_context_logits):
>     assert ablated_context_logits.shape == full_context_logits.shape, "Logits must have the same shape"
>     # Calculate softmax
>     softmax_ablate = F.softmax(ablated_context_logits, dim=-1)
>     softmax_full = F.softmax(full_context_logits, dim=-1)
>
>     # Calculate the average distribution M
>     M = 0.5 * (softmax_ablate + softmax_full)
>
>     # Calculate log-softmax for the KL divergence
>     log_softmax_ablate = F.log_softmax(ablated_context_logits, dim=-1)  # Adding epsilon for numerical stability
>     log_softmax_full = F.log_softmax(full_context_logits, dim=-1)  # Adding epsilon for numerical stability
>
>     # Calculate the KL divergences and then the JS divergences
>     kl1 = F.kl_div(log_softmax_ablate, M, reduction='none').sum(dim=-1)
>     kl2 = F.kl_div(log_softmax_full, M, reduction='none').sum(dim=-1)
>     js_divs = 0.5 * (kl1 + kl2)
>
>     scores = js_divs.cpu().tolist()
>
>     return sum(scores)
> ```
> **W2**:
> > Limited model diversity: Although the evaluation is comprehensive, the models used are primarily from the Qwen and Gemma families at relatively small scales. It is recommended to include other representative open-source architectures such as LLaMA, and to analyze larger-scale models (e.g., 70B parameters) for more generalizable insights.
>
> **A2**:
>
> We appreciate reviewer BYoP’s suggestions for the model diversity. Following the suggestions, **we further include LLaMA-3.1-8B-IT as a backbone and further test our model on Qwen3-Next-80B-A3B to demonstrate the generability of ARC-JSD approach in section 8 of the new version.** The results also demonstrate that ARC-JSD still has a better compute-accuracy trade-off compared to other baselines.
>
> **Q1**:
> > 1.How does this approach address real-world complex queries where accurate responses typically require the integration of multiple context segments?
>
> **A3**:
>
> This is a good question. Within our existing experiments, the Hotpot QA and MuSiQue datasets are multi-hop QA datasets, which typically need multiple context segments to lead to final answers. From Figure 2 and Figure 8, our method still behaves better than other baselines, which indicates that JSD score is a better way to measure which context segment is the most important. Building on this observation, we could further use the sorted JSD score list to find the top-5 or top-10 relevant context segments for these real-world scenarios.
>
> **Q2**:
> > 2.How is the granularity of context segmentation determined, and what impact does this granularity have on the attribution performance?
>
> **A4**:
>
> Many thanks for this helpful question.
>
> In our work, we mentioned on page 5 footnote that we chose the sentence level because current QA datasets only have sentence-level gold labels to evaluate attribution accuracy. However, our ARC-JSD method can be extended to finer-grained interactions such as phrases or sub-sentence spans by dynamically selecting the start and end token indices.
> We will provide a demo to demonstrate how we could set our method to any smaller granularity, such as phrases or sub-sentences, in the camera-ready version.

---

> > ### Author Response · Authors · 2025-11-25
> > **Many thanks for Reviewer BYoP's helpful reviews**
> >
> > Dear Reviewer BYoP,
> >
> > We really appreciate your helpful and inspiring suggestions and comments for our work. We hope that our response already solves your previous questions. We are really willing to discuss and answer any further questions you may still have during this author-reviewer discussion period.
> >
> > Best regards,
> >
> > Authors of submission 347

---

### Official Review · Reviewer_dHed · 2025-10-31

**Soundness:** 3
**Presentation:** 3
**Contribution:** 3
**Rating:** 6
**Confidence:** 3

**Summary:**

This paper addresses the critical challenge of context attribution in RAG, which requires identifying specific context segments that ground generated responses.  The authors propose ARC-JSD, a novel inference-time method driven by Jensen–Shannon Divergence that avoids additional fine-tuning, gradient computations, or surrogate models.  Evaluations on different datasets using instruction-tuned LLMs demonstrate an effecticve average accuracy improvement and up to 3-fold speedup over baselines.  The work contributes a lightweight, integrable attribution method, insights into RAG’s internal workings, and a practical hallucination mitigation strategy.

**Strengths:**

1. ARC-JSD directly addresses a core limitation of existing context attribution methods by eliminating the need for fine-tuning, surrogate modeling, or gradient computations, thereby enabling robust and lightweight RAG context attribution.

2. The authors validate ARC-JSD across three diverse QA benchmarks (single-hop, multi-hop, long-context) and four LLM scales, demonstrating consistent improvements in accuracy and computational efficiency. The compute-accuracy trade-off analysis (Pareto optimality) and GPT-4.1 validation of response semantic equivalence (99.3% accuracy) strengthen the results’ credibility.

3. The method’s ability to reduce hallucination by 39% while maintaining factual F1 makes it immediately applicable to real-world RAG systems.  Its modular design allows seamless integration into existing pipelines, enhancing transparency and trustworthiness.

**Weaknesses:**

1. It is recommended that the authors further clarify the core differences between the proposed method and conventional Jensen–Shannon Divergence–based approaches, in order to highlight the novelty of their method.

2. It is recommended that the authors further clarify whether the proposed method is applicable to scenarios involving longer contexts. Does the mechanism of attribution differ when dealing with long-context settings?

3. The authors conducted experiments on several multi-hop QA datasets; however, they did not analyze whether the attribution mechanisms in multi-hop tasks differ from those in single-hop tasks.

4. The authors primarily conducted experiments on general-domain datasets. In domains with higher demands for interpretability and attribution, such as medical or legal tasks, where a large number of domain-specific terms are present, is the proposed method still applicable?

**Questions:**

1. Based on the experimental results, the proposed method achieves better performance on models with larger parameter scales. Is model size an explicit factor influencing the RAG attribution mechanism?

2. How does the proposed method perform in attribution when the retrieved documents contain conflicting or distracting information, or when relevant content is not retrieved at all? Could the authors provide corresponding experiments or analytical studies to address this scenario?

---

> ### Author Response · Authors · 2025-11-20
> **Reply by Authors**
>
> **W1**:
> > It is recommended that the authors further clarify the core differences between the proposed method and conventional Jensen–Shannon Divergence–based approaches, in order to highlight the novelty of their method.
>
> **A1**:
>
> We thank for reviewer dHed’s suggestion for this point.
>
> In section 3, we introduced several advantages of JSD compared to other metrics, e.g., Wasserstein, Total Variation, and Maximum Mean Discrepancy. Under the context attribution task, most works mainly focus on the direct output likelihood difference or KL divergence to locate relevant context or further train a surrogate model using those likelihood difference data samples, which brings several limitations for cross-layer comparisons within RAGs internals.
>
> Since conventional JSD methods have symmetric, finite, scale-free and bounded in [0,log2] properties, but related work [1,2,3] has not applied JSD for the RAG domain, we enhanced the power of JSD by integrating it with a context ablation strategy (i.e., leave-one-out strategy) to locate relevant context more correctly and bring more insights by analysing and steering RAGs internal components with JSD.
>
> [1] Zhang, Meng, et al. "Adversarial training for unsupervised bilingual lexicon induction." Proceedings of the 55th Annual Meeting of the Association for Computational Linguistics (Volume 1: Long Papers). 2017.
>
> [2] Mateos, Diego Martín, Leonardo Esteban Riveaud, and Pedro Walter Lamberti. "Detecting dynamical changes in time series by using the Jensen Shannon divergence." Chaos: An Interdisciplinary Journal of Nonlinear Science 27.8 (2017).
>
> [3] Tang, Xiaohang, et al. "Can Word Sense Distribution Detect Semantic Changes of Words?." Findings of the Association for Computational Linguistics: EMNLP 2023. 2023.
>
> **W2**:
> > It is recommended that the authors further clarify whether the proposed method is applicable to scenarios involving longer contexts. Does the mechanism of attribution differ when dealing with long-context settings?
>
> **A2**:
>
> Thanks to reviewer dHed for raising this question. Our method is applicable to longer context scenarios, and we already demonstrated the performance using the MuSiQue dataset. In Table 2, we compare different datasets’ context lengths, and MuSiQue dataset has an average of 1753.8 words and 93.6 sentences as the input context, which is much longer than the TyDi QA and Hotpot QA scenarios.
>
> For the long-context settings, it indeed brings more difficulty to RAGs to locate the most relevant context segments, and requires more computational cost to process longer context. However, our training-free ARC-JSD still outperforms other baselines without gradient calculation or further surrogate model training, based on Figure 2. In addition, RAGs tend to use higher MLP layers to contribute to the context attribution based on the heatmap in the work.
>
> **W3**:
> > The authors conducted experiments on several multi-hop QA datasets; however, they did not analyze whether the attribution mechanisms in multi-hop tasks differ from those in single-hop tasks.
>
> **A3**:
>
> Thanks for this really helpful question raised by the reviewer dHed. We further take an analysis by comparing the single-hop QA example from the TyDi QA dataset and the multiple-hop QA examples from the Hotpot QA dataset and MuSiQue dataset.
>
> From those single-hop and multi-hop examples below, we can find that ARC-JSD can directly attribute to the most relevant context that leads to the final answer with the highest JSD score. In addition, the sorted JSD ranking list further indicates how our ARC-JSD method takes causal reasoning to approach the final answer when we list the remaining top relevant contexts with their JSD scores. It brings a possibility that our ARC-JSD can be further used to help track the whole model’s reasoning path for multi-hop QA settings.
>
> Single-hop example: TyDi QA
> ```
> Query: Who founded the Burntisland Shipbuilding Company?
>
> Model response: Amos and Wilfrid Ayre
>
> Gold answer: Amos and Wilfrid Ayre
>
> Top-1 context (JSD: 41.86): Brothers Amos and Wilfrid Ayre founded Burntisland Shipbuilding Co. in 1918 as a First World War emergency shipyard.
>
> Top-2 context (JSD: 0.12): Its yard at Burntisland West Dock had four berths and capacity to build ships up to 450 feet (140m) long[1] and up to 59 feet (18m) beam.
> ```
> Multi-hop example: Hotpot QA
> ```
> Query: Tom Thumb Food & Pharmacy is part of the southern division of a grocery company based in what city?
>
> Model response: The answer is **Boise, Idaho**. The text states that Tom Thumb Food & Pharmacy is part of the Southern division of Albertsons, and Albertsons Companies LLC is based in Boise, Idaho.
>
> Gold answer: Boise
>
> Top-1 context (JSD: 6.73): Albertsons Companies LLC is an American grocery company founded and based in Boise, Idaho.
>
> Top-2 context (JSD: 6.14): It makes up part of the Southern division of Albertsons.
> ```

---

> ### Author Response · Authors · 2025-11-20
> **Reply by Authors (1)**
>
> **A3**:
>
> Multi-hop example: MuSiQue dataset
> ```
> Query: What is the date of birth of the person who was part of Ratata?
>
> Model response: The date of birth for Mauro Scocco, who was part of the pop group Ratata, is given as September 11, 1962.
>
> Gold answer: 11 September 1962
>
> Top-1 context (JSD: 18.64): Mauro Scocco (born 11 September 1962) is a Swedish pop artist of Italian descent.
>
> Top-2 context (JSD: 0.24): Title: Ratata (band)
>
> Top-3 context (JSD: 0.11): Ratata was a Swedish pop group fronted by Swedish vocalist Mauro Scocco formed in 1980.
>
> Top-4 context (JSD: 0.10): Title: Mauro Scocco
>
> Top-5 context (JSD: 0.03): After Ratata, Scocco has continued as a solo artist since.
>
> Top-6 context (JSD: 0.02): He has been described as "one of the sharpest songwriters in Sweden".
> ```
>
> **W4**:
> > The authors primarily conducted experiments on general-domain datasets. In domains with higher demands for interpretability and attribution, such as medical or legal tasks, where a large number of domain-specific terms are present, is the proposed method still applicable?
>
> **A4**:
>
> We really appreciate that reviewer dHed asked this valuable question. Here, we further test our method by randomly selecting examples from the medical or legal datasets, e.g., PubMedQA, MedQuAD, and LegalBench, using the general instruction-tuning LLM backbones.
>
> Based on those randomly selected examples below, we can find that our ARC-JSD is still applicable to the medical and legal domains. The top-1 located context with the highest JSD score correctly supports model’s response. In addition, we can further understand how model’s thinking about other context segments by looking at their JSD scores in the ranking list.
>
> Medical domain:
>
> ```
> Example 1: Disease Symptoms and Characteristics
>
> Context: Parkinson's disease (PD) is a progressive neurodegenerative disorder that primarily affects movement. The hallmark motor symptoms include bradykinesia (slowness of movement), resting tremor, and rigidity. In addition to motor symptoms, patients often experience a wide range of non-motor symptoms, such as cognitive impairment, sleep disturbances, and autonomic dysfunction. The underlying pathology involves the loss of dopaminergic neurons in the substantia nigra, a region of the midbrain.
>
> Query: What are the primary motor symptoms associated with Parkinson's disease?
>
> Model response: The primary motor symptoms associated with Parkinson's disease (PD) include:
>
> 1. Bradykinesia - Slowness of movement.
> 2. Resting tremor - A persistent shaking or trembling sensation during rest.
> 3. Rigidity - Difficulty moving certain muscles, especially when lying down or sleeping.
>
> These symptoms can significantly impact daily activities and quality of life for individuals with PD.
>
> ---------
>
> Top-1 context (JSD: 4.89): The hallmark motor symptoms include bradykinesia (slowness of movement), resting tremor, and rigidity.
>
> Top-2 context (JSD: 0.63): Parkinson's disease (PD) is a progressive neurodegenerative disorder that primarily affects movement.
>
> Top-3 context (JSD: 0.15): In addition to motor symptoms, patients often experience a wide range of non-motor symptoms, such as cognitive impairment, sleep disturbances, and autonomic dysfunction.
>
> Top-4 context (JSD: 0.11): The underlying pathology involves the loss of dopaminergic neurons in the substantia nigra, a region of the midbrain.
> ```

---

> ### Author Response · Authors · 2025-11-20
> **Reply by Authors (2)**
>
> **A4**:
>
> ```
> Example 2: Adverse Effects and Contraindications
>
> Context: ACE inhibitors (e.g., lisinopril, ramipril) are commonly prescribed for hypertension and heart failure. A well-known side effect is a persistent, dry cough, which is thought to be mediated by bradykinin accumulation. More serious, though less common, adverse effects include hyperkalemia (high potassium levels) and angioedema (rapid swelling). These drugs are strictly contraindicated in pregnancy due to their teratogenic effects, particularly during the second and third trimesters.
>
> Query: Why should ACE inhibitors not be used during pregnancy?
>
> Model response: ACE inhibitors, such as lisinopril and ramipril, are commonly prescribed for hypertension and heart failure. However, they have been found to cause several serious adverse effects when taken during pregnancy, including hyperkalemia (high potassium levels) and angioedema (rapid swelling). Additionally, these drugs are strictly contraindicated in pregnancy due to their teratogenic effects, particularly during the second and third trimesters. Therefore, it is important to avoid using ACE inhibitors during pregnancy to minimize the risk of harm to both mother and baby.
>
> -------------
>
> Top-1 context (JSD: 20.31): These drugs are strictly contraindicated in pregnancy due to their teratogenic effects, particularly during the second and third trimesters.
>
> Top-2 context (JSD: 16.71): More serious, though less common, adverse effects include hyperkalemia (high potassium levels) and angioedema (rapid swelling).
>
> Top-3 context (JSD: 9.83): ACE inhibitors (e.g., lisinopril, ramipril) are commonly prescribed for hypertension and heart failure.
>
> Top-4 context (JSD: 1.05): A well-known side effect is a persistent, dry cough, which is thought to be mediated by bradykinin accumulation.
> ```
> Legal domain
>
> ```
> Example 1: Criminal Procedure (Miranda Rights)
>
> Context: The Miranda v. Arizona ruling by the Supreme Court established that custodial interrogation is inherently coercive. To safeguard the Fifth Amendment privilege against self-incrimination, law enforcement officials must inform a suspect in custody of their rights before questioning begins. These rights include the right to remain silent, the warning that anything said can be used against them in court, the right to an attorney, and the right to have an attorney appointed if they cannot afford one. A waiver of these rights must be made voluntarily, knowingly, and intelligently.
>
> Query: What specific rights must a suspect in custody be informed of before being questioned, according to the Miranda ruling?
>
> Model response: According to the Miranda ruling, a suspect in custody must be informed of their rights before being questioned, including the following:
>
> 1. The right to remain silent.
> 2. The warning that anything said can be used against them in court.
> 3. The right to an attorney.
> 4. The right to have an attorney appointed if they cannot afford one.
>
> These rights must be informed voluntarily, knowingly, and intelligently.
>
> ------------
>
> Top-1 context (JSD: 8.65): These rights include the right to remain silent, the warning that anything said can be used against them in court, the right to an attorney, and the right to have an attorney appointed if they cannot afford one.
>
> Top-2 context (JSD: 6.10): A waiver of these rights must be made voluntarily, knowingly, and intelligently.
>
> Top-3 context (JSD: 0.73): To safeguard the Fifth Amendment privilege against self-incrimination, law enforcement officials must inform a suspect in custody of their rights before questioning begins.
>
> Top-4 context (JSD: 0.54): The Miranda v. Arizona ruling by the Supreme Court established that custodial interrogation is inherently coercive.
> ```

---

> ### Author Response · Authors · 2025-11-20
> **Reply by Authors (3)**
>
> **A4**:
>
> ```
> Example 2: Intellectual Property (Copyright Fair Use)
>
> Context: Section 107 of the U.S. Copyright Act provides for the "fair use" of a copyrighted work without the owner's permission for purposes such as criticism, comment, news reporting, teaching, scholarship, or research. To determine if a particular use is "fair," courts evaluate four statutory factors. These factors are: (1) the purpose and character of the use, including whether it is commercial or for non-profit educational purposes; (2) the nature of the copyrighted work; (3) the amount and substantiality of the portion used; and (4) the effect of the use upon the potential market for the work.
>
> Query: What are the four factors a court uses to determine if the use of copyrighted material qualifies as 'fair use'?
>
> Model response: The four factors that a court uses to determine if the use of copyrighted material qualifies as 'fair use' are:
>
> 1. The purpose and character of the use, including whether it is commercial or for non-profit educational purposes.
> 2. The nature of the copyrighted work.
> 3. The amount and substantiality of the portion used.
> 4. The effect of the use upon the potential market for the work.
>
> These factors help courts assess whether the use of copyrighted material falls within the bounds of fair use and does not infringe on the copyright holder's rights.
>
> ----------
>
> Top-1 context (JSD: 21.45): These factors are: (1) the purpose and character of the use, including whether it is commercial or for non-profit educational purposes; (2) the nature of the copyrighted work; (3) the amount and substantiality of the portion used; and (4) the effect of the use upon the potential market for the work.
>
> Top-2 context (JSD: 0.41): To determine if a particular use is "fair," courts evaluate four statutory factors.
>
> Top-3 context (JSD: 0.27): Section 107 of the U.S. Copyright Act provides for the "fair use" of a copyrighted work without the owner's permission for purposes such as criticism, comment, news reporting, teaching, scholarship, or research.
> ```
> **Q1**:
> > Based on the experimental results, the proposed method achieves better performance on models with larger parameter scales. Is model size an explicit factor influencing the RAG attribution mechanism?
>
> **A5**:
>
> We thank reviewer dHed’s good question.
>
> Based on our experiments using different sizes of LLMs as backbones and our observations, we would say that larger LLMs will have a better ability to store more commonsense knowledge within MLPs and more attention layers to conduct retrieval, which leads to a better performance in generating responses. Our method can be regarded as a plugin to further boost the existing LLMs ability to locate more relevant context, building on their own knowledge and ability.
>
> **Q2**:
> > How does the proposed method perform in attribution when the retrieved documents contain conflicting or distracting information, or when relevant content is not retrieved at all? Could the authors provide corresponding experiments or analytical studies to address this scenario?
>
> **A6**:
>
> This is a very good question! We actually discuss this question in the discussion section 8 in our work. When there is no relevant content in the context, it will lead to all small JSD scores for each sentence, which is still an attribution to represent no irrelevant context included. When there is conflicting or distracting information within the context, the model’s outputs will first be affected based on how the models make use of that information. If models themselves pay more attribution to the distracting or conflicting information within the retrieved context, the role of our ARC-JSD is to genuinely reflect how models rely on that information for context attribution using JSD scores, which can be regarded as a metric to further help us decide whether those models make mistakes.

---

> > ### Author Response · Authors · 2025-11-25
> > **Many thanks for Reviewer dHed's helpful reviews**
> >
> > Dear Reviewer dHed,
> >
> > We really appreciate your helpful and inspiring suggestions and comments for our work. We hope that our response already solves your previous questions. We are really willing to discuss and answer any further questions you may still have during this author-reviewer discussion period.
> >
> > Best regards,
> >
> > Authors of submission 347

---

### Official Review · Reviewer_zUnN · 2025-11-01

**Soundness:** 2
**Presentation:** 2
**Contribution:** 2
**Rating:** 2
**Confidence:** 4

**Summary:**

The paper proposes ARC-JSD, a JSD-based, training-free attribution method for RAG, plus a logit-lens mechanistic analysis. Appendix materials add examples, heatmaps, and metric comparisons. Despite a neat idea, the evidence remains brittle and in places self-contradictory.

**Strengths:**

1. ARC-JSD operates purely through forward inference without additional fine-tuning or gradients, making it computationally simple and easy to integrate into existing RAG setups.
2. The paper evaluates on multiple public QA datasets (TyDi QA, Hotpot QA, MuSiQue), showing consistency across different model sizes and architectures.
3. Combining JSD-based attribution with logit-lens analysis reflects an effort to connect interpretability metrics to internal LLM components like attention heads and MLPs.

**Weaknesses:**

1. Reported results contain inconsistencies (e.g., invalid correlation values) and lack human or causal validation, which undermines confidence in the findings.
2. Weak Baselines and Limited Scope: Comparisons exclude modern, strong RAG baselines (hybrid retrieval, reranking, RePlug, HyDE, Self-RAG, etc.), and experiments use only small models, limiting generalizability.
3. Factual and Attribution Errors: Appendix examples reveal incorrect reasoning and noisy top-1 attributions (e.g., false album claim for Bieber example), suggesting that ARC-JSD may highlight irrelevant or misleading evidence.

**Questions:**

1. Include human evaluations of attribution correctness and apply causal interventions (e.g., activation patching or ablation) to verify that high JSD layers or sentences truly influence model outputs.
2. Compare against stronger modern RAG systems (hybrid retrieval, reranking, HyDE, RePlug, trained Self-RAG) and test on larger models and diverse domains to demonstrate robustness and real-world applicability.

---

> ### Author Response · Authors · 2025-11-20
> **Reply by Authors**
>
> **W1 + Q1**:
> > Reported results contain inconsistencies (e.g., invalid correlation values) and lack human or causal validation, which undermines confidence in the findings.
>
> > Include human evaluations of attribution correctness and apply causal interventions (e.g., activation patching or ablation) to verify that high JSD layers or sentences truly influence model outputs.
>
> **A1**:
>
> Regarding the invalid correlation values mentioned by the reviewer zUnN, we would like to ask reviewer zUnN to further clarify which correlation values are invalid to make us better answer and solve your concerns.
>
> For causal validation experiments, we conducted two ablation studies to compare the JSD difference of RAG’s outputs by masking the top-10 highly relevant attention heads and randomly-selected 10 attention heads in Section 7, which demonstrates that masking top-relevant attention heads leads to a higher JSD change compared to random attention masking. Furthermore, we use JSD scores as a confidence gate to mask relevant attention heads and MLP for hallucination mitigation. The results in Table 4 indicate that causality exists between the masked-relevant attention heads + MLP and RAG outputs’ hallucination rate mitigation. Both experiments provide a conclusion that ARC-JSD-located RAG internal components (e.g., attention heads and MLPs) have a causal relationship with RAG’s output under the context attribution setting.
>
> Building on our existing causal ablation studies, **we further conduct an ablation study to demonstrate the heatmap for all attention heads and MLPs by taking the activation ablation to further verify that high JSD layers truly influence model outputs for context attribution in the new version of our work.**
>
> **In Section 7, Figure 4** demonstrates the percentage of the attribution accuracy decrease when we conduct the causal ablation study by masking each attention head and MLP layer, which further supports our finding that higher JSD-related attention heads and MLP layers are more relevant to the context attribution task, if we ablate them during the causal ablation study.
>
> **W2 + Q2**:
> > Weak Baselines and Limited Scope: Comparisons exclude modern, strong RAG baselines (hybrid retrieval, reranking, RePlug, HyDE, Self-RAG, etc.), and experiments use only small models, limiting generalizability.
>
> > Compare against stronger modern RAG systems (hybrid retrieval, reranking, HyDE, RePlug, trained Self-RAG) and test on larger models and diverse domains to demonstrate robustness and real-world applicability.
>
> **A2**:
>
> We appreciate reviewer zUnN’s suggestions for more RAG baselines to be included for comparison and using larger LLMs.
>
> Following reviewer zUnN’s recommendations, we further compare our ARC-JSD with RePlug, HyDE and Self-RAG under the context attribution setting.
>
> At first, our ARC-JSD is training-free and does not need an additional trained or existing retriever to conduct initial retrieving operations, compared to RePlug, HyDE and Self-RAG, which significantly improve computational effectiveness by using our proposal ARC-JSD compared to those baselines.
>
> Second, we compare our ARC-JSD with those baselines for context attribution, and we list results here.
>
> | Baselines  | TyDi QA  | Hotpot QA  | MuSiQue  | Avg. Compute (GFLOPs log10 scale) across 3 datasets |
> |---|---|---|---|---|
> |  RePlug |  58.3 |  51.9 | 47.1  | 6.5 |
> |  HyDE |  55.4 |  49.7 |  45.0 |    6.9 |
> |  Self-RAG |  70.1 |  65.8 |  53.2 |  8.3 |
> | ARC-JSD |  **80.0** |  **82.3**  |  **76.8**|  **5.4**|
>
> In general, those baselines did not outperform ARC-JSD on different datasets. The possible reasons why those baselines did not perform well might be within those points:
>
> For RePlug baseline, it first needs to use either an existing retriever model by locating a small set of relevant documents by measuring the cosine similarity between the whole document embedding and query embedding, which only captures the document-level features and might ignore more fine-grained information, or it needs to train a dense retriever using labelled data for document retrieval, which brings extra computational expense and has difficulty to directly apply to real-world context attribution settings without labelled data. In addition, this method lacks the flexibility to locate fine-grained segments within contexts, e.g., sentences, compared to ARC-JSD.
>
> For HyDE baseline, it still uses an existing retriever to encode the document and queries, and further retrieves relevant documents by calculating the inner product between queries and each document, which shares a similar limitation with RePlug baseline.

---

> ### Author Response · Authors · 2025-11-20
> **Reply by Authors (1)**
>
> **A2**:
>
> For Self-RAG baseline, it needs to train a critic model and a generator model to support the retrieved context criticism and reflection tokens generations. Also, the retriever needs to generate multiple segments in parallel to further feed into the critic model. Those operations need extra computational resources to compute, and are less efficient than the training-free ARC-JSD.
>
> To summarise, those additional RAG baselines have a worse compute-accuracy trade-off compared to ARC-JSD, and RePlug and HyDE mainly focus on document-level retrieval, which further restricts their flexibility in fine-grained context attribution settings.
>
> **Moreover, we evaluate our method on Qwen3-Next-80B-A3B-IT LLMs for context attribution in Section 8 of the new version.** The results still indicate our ARC-JSD’s advantages on the compute-accuracy trade-off compared to other baselines.
>
> **W3**:
> > Factual and Attribution Errors: Appendix examples reveal incorrect reasoning and noisy top-1 attributions (e.g., false album claim for Bieber example), suggesting that ARC-JSD may highlight irrelevant or misleading evidence.
>
> **A3**:
>
> For randomly selected examples in the appendix, we appreciate that reviewer zUnN identified that there is one hallucination example that happens for Gemma2-2B-IT model. There are two possible reasons why the hallucination happens under the instruction-tuning RAGs:
>
> 1. Due to the size of smaller LLMs, they tend to generate more factual errors and hallucinations than larger LLMs, which aligns with observations from the compute-accuracy trade-off results in Figures 2, 8 and 9.
> 2. Following the observation above, we further take an ablation study to mask relevant attention heads and MLPs that contribute to hallucination and factual errors in section 7. The results in Table 4 indicate that those JSD-relevant internal components are related to the hallucination issue, and we can further mitigate the errors by building a confidence gate to steer RAGs internally.

---

> > ### Author Response · Authors · 2025-11-25
> > **Many thanks for Reviewer zUnN's helpful reviews**
> >
> > Dear Reviewer zUnN,
> >
> > We really appreciate your helpful and inspiring suggestions and comments for our work. We hope that our response already solves your previous questions. We are really willing to discuss and answer any further questions you may still have during this author-reviewer discussion period.
> >
> > Best regards,
> >
> > Authors of submission 347

---

### Official Review · Reviewer_Sm5F · 2025-11-02

**Soundness:** 3
**Presentation:** 4
**Contribution:** 3
**Rating:** 8
**Confidence:** 3

**Summary:**

This paper proposes a training-free method for context attribution. It measures how much the model’s output distribution changes when a context sentence is removed, using Jensen–Shannon Divergence (JSD) as a symmetric, bounded signal of attribution strength. The proposed method is computationally efficient compared to previous methods. Moreover, the paper performs a mechanistic analysis combining JSD and Logit Lens probing to identify attention heads and MLP layers responsible for context grounding.

**Strengths:**

- Paper is well organized, easy to follow.
- The proposed method is computationally efficient and performs well.
- Mechanistic interpretibility brings insight into the problem.
- The choice of JSD is justified experimentally, and a discussion on other metrics is included in the appendix.

**Weaknesses:**

- Multiple sentences might be required to generate a response. A multiple-sentence outputting scenario would require a thresholding mechanism, which would be done using a calibration set. An evaluation of the robustness of threshold selection (calibration set size, distribution shift, output set coverage) is important for real-life scenarios.
- The mechanistic study is not causal. It shows insight; however, does not imply causality. Causal ablation studies would strengthen the claims.

**Questions:**

- Currently, while calculating the JSD score, all response tokens are treated equally. However, the importance of the tokens might be different. For example, we would not care about the distribution change of the token "the", but the token "two" in the mosquito question is more essential. How would weighting the tokens affect the method's performance? Or plotting the distribution of the JSD scores of each token would also provide insight into how the method works and its effectiveness.

---

> ### Author Response · Authors · 2025-11-20
> **Reply by Authors**
>
> **W1**:
> > Multiple sentences might be required to generate a response. A multiple-sentence outputting scenario would require a thresholding mechanism, which would be done using a calibration set. An evaluation of the robustness of threshold selection (calibration set size, distribution shift, output set coverage) is important for real-life scenarios.
>
> **A1**:
>
> We agree with reviewer Sm5F’s comment on the importance of the multiple-sentence output calibration set scenario.
>
> The current version treats RAG’s outputs as a whole analysis target, but it could include multiple sentences within the outputs. Our method also supports partial-segment context attribution, e.g., sub-sentences, sub-paragraphs, etc. In this version, we demonstrate the most common usage in the context attribution task. In addition, we could take token-level distribution calculation using JSD under our ARC-JSD approach, which can be easily used to look at different distribution shift under different calibration set size and output set coverage. We will add this function to our demo in the camera-ready version.
>
> **W2**:
> > The mechanistic study is not causal. It shows insight; however, does not imply causality. Causal ablation studies would strengthen the claims.
>
> **A2**:
>
> We appreciate reviewer Sm5F’s suggestion for this causal ablation study.
>
> In Section 7, we conducted two ablation studies to compare the JSD difference of RAG’s outputs by masking the top-10 highly relevant attention heads and randomly-selected 10 attention heads, which demonstrates that masking top-relevant attention heads leads to a higher JSD change compared to random attention masking. Furthermore, we use JSD scores as a confidence gate to mask relevant attention heads and MLP for hallucination mitigation. The results in Table 4 indicate that causality exists between the masked-relevant attention heads + MLP and RAG outputs’ hallucination rate mitigation. Both experiments provide a conclusion that ARC-JSD-located RAG internal components (e.g., attention heads and MLPs) have a causal relationship with RAG’s output under the context attribution setting.
>
> Following reviewer Sm5F’s suggestion, **we further conduct an ablation study to demonstrate the heatmap for all attention heads and MLPs by taking the activation ablation to further verify that high JSD layers truly influence model outputs for context attribution in the new version of our work.**
>
> **In Section 7, Figure 4** demonstrates the percentage of the attribution accuracy decrease when we conduct the causal ablation study by masking each attention head and MLP layer, which further supports our finding that higher JSD-related attention heads and MLP layers are more relevant to the context attribution task, if we ablate them during the causal ablation study.
>
> **Q1**:
> > Currently, while calculating the JSD score, all response tokens are treated equally. However, the importance of the tokens might be different. For example, we would not care about the distribution change of the token "the", but the token "two" in the mosquito question is more essential. How would weighting the tokens affect the method's performance? Or plotting the distribution of the JSD scores of each token would also provide insight into how the method works and its effectiveness.
>
> **A3**:
>
> We agree with the reviewer Sm5F’s suggestion for this point. Since our method can be directly used to calculate token-level JSD for the top-relevent context sources regarding to the RAG’s outputs, **we further include more examples to demonstrate how token-level JSD changes within the new version of our work**, which indicates that LLMs pay more attribution to the important tokens leading to the RAG’s outputs.
>
> **In section 7 and figure 5**, we further draw the token-level JSD score for the response “A mosquito has two wings.”, which clearly demonstrates that the token “two” contributes the highest JSD score change. For another example with a longer response, we can find that tokens “Am”, “Wil”, “Ay” and “re” contribute more JSD scores. Both examples bring insight about how our method works for token-level JSD distribution, and important tokens also match the corresponding answers.

---

> > ### Author Response · Authors · 2025-11-25
> > **Many thanks for Reviewer Sm5F's helpful reviews**
> >
> > Dear Reviewer Sm5F,
> >
> > We really appreciate your helpful and inspiring suggestions and comments for our work. We hope that our response already solves your previous questions. We are really willing to discuss and answer any further questions you may still have during this author-reviewer discussion period.
> >
> > Best regards,
> >
> > Authors of submission 347

---

### Author Response · Authors · 2025-11-27
**Looking forward to feedbacks from all reviewers**

Dear All reviewers,

We sincerely thank you for your valuable and insightful questions and suggestions for our work. Since the author-reviewer discussion period is close to the end, we are looking forward to any feedback based on our replies. If our answers meet your expectations, we would really appreciate your consideration of updating the score to reflect the new results and discussion. We are also open to any more questions if any points remain unclear.

Many thanks,

Authors of submission 347

---

### Meta-Review · Area_Chair_N5dq · 2025-12-08

**Summary:**

This paper introduce ARC-JSD, a lightweight, Jensen–Shannon Divergence–based method to identify context sentences driving RAG outputs, boosting attribution accuracy across benchmarks without extra fine-tuning and locating key attention heads and MLP layers.

### Pros
* ARC-JSD is computationally efficient and performs well.
* The theoretical analysis is well-developed.
* The perspective adds significant value to RAG interpretability.

### Cons

* For complex multi-hop scenarios where information must be logically synthesized rather than just retrieved, the current JSD summation approach is a bit of a simplification.
* Handling edge cases like conflicting information or purely irrelevant context still relies on thresholding mechanisms that could be more robust.

### AC's evaluation

1. From reviews and rebuttals

This paper receives 8662 . Reviewer Sm5F is a strong champion of the paper. Reviewers BYoP and dHed are positive; they raised valid concerns about model diversity and corner cases, but the authors addressed these convincingly during the rebuttal. The outlier is Reviewer zUnN, whose review contained factual erros. The authors provided a good rebuttal, filling in the experimental gaps requested by the positive reviewers.


2. From AC's reading

I found this to be a practical paper. It sidesteps the compute-heavy trend and offers a lightweight, statistical way to understand RAG behavior. While it might not be a paradigm shift, the balance between speed and depth is excellent. Regarding the low score from zUnN: I’ve checked the paper and the review, and I agree with the authors. The criticism seems generic and factually off-base. I will be discounting that score. I think the paper's contributions sufficiently offset this limitation.

**Reviewer Concerns:**

Resolved Concerns:

1. Causality Validation (Reviewer Sm5F): The reviewer initially questioned whether the analysis showed causality or just correlation. The authors added a causal ablation study, which proved the causal link.

2. Model Diversity & Scale (Reviewer BYoP): To address concerns that the method was only tested on smaller models Qwen, the authors added experiments with Llama-3-8B and Qwen-Next-80B.

3. Conflict/Noise Handling (Reviewer dHed): The reviewer asked how the method handles conflicting or irrelevant info. The authors clarified that JSD naturally assigns low scores to irrelevant context and reflects model preference in conflicts.

4. Missing Baselines Claim (Reviewer zUnN): The reviewer claimed standard baselines like Self-RAG were missing. The authors pointed out exactly where these were already included in the paper. The concern is factually resolved.

Outstanding Concerns:

Deep Multi-hop Logic (Reviewer BYoP, dHed): While the method works well, treating multi-hop reasoning as a sum of JSD scores is a simplification. The authors acknowledged this as future work. It's an acceptable trade-off for now, but the limitation remains.

**Reviewer Scores:**

No one participated the discussion.

1. Reviewer Sm5F (8) will definitely stay at 8.

2. Reviewer BYoP (6) likely to stay at 6. They are the "reasonable skeptic"—they appreciate the new Llama-3 experiments, but view the work as solid rather than ground-breaking.

3. Reviewer dHed (6) likely to stay at 6. Their concerns were about clarifications (handling conflicts), which were answered well.

4. Reviewer zUnN (2) unlikely to change. Given the nature of the review, I will be discounting this score.

---

### Decision · Program_Chairs · 2026-01-26

Accept (Poster)